# InfoGlobe: Local-and-Global Information-Preserving Statistical Manifold Learning for Single-Cell Transcriptomics

Cheng Wang [* 1]   Jinpu Cai [* 2]   Chongxiao Mao [* 1]   Yuxuan Wang [1]   Xinzhu Jiang [2]   Yunhao Qiao [1]   Luqi Yang [1]   Luting Zhou [2]   Qiuyu Lian [† 3 4]   Hongyi Xin [† 1 5]

## Abstract

Geometry-preserving dimension reduction is critical for single-cell transcriptomics, where low-dimensional distances should reflect biological divergence between cell types along the transcriptomic manifold. Due to inadequate metrics, the global structure is not sufficiently preserved in the low-dimensional manifold in standard dimension reduction regimes. We model RNA counts as Multinomial samples, leveraging their nested closure property: gene-level counts refine functional gene-group counts via nested Multinomial distributions. Extending Chentsov's Theorem, we show that the Fisher-Rao metric on coarse (gene-group) and fine (gene) statistical manifolds is isometric. Following this isometry property, we propose InfoGlobe, an information-preserving statistical manifold learning framework that projects cells from high-dimensional hyperspheres (full transcriptome) to low-dimensional hyperspheres (functional groups) while preserving information geometry. Embeddings on the low-dimensional sphere explicitly represent Multinomial distributions by functional gene groups. Benchmarks demonstrate superior preservation of local-and-global cell-type geodesic distances, automatic and robust gene-group discovery, nuanced cell subtype resolution without manual feature engineering and natural batch effect mitigation without explicit alignments.

## 1. Introduction

Geometry-preserving dimension reduction has become a central tool in single-cell transcriptomics (Wolf et al., 2018; Luecken & Theis, 2019), where it underlies our ability to reconstruct and interpret the transcriptomic landscape of complex tissues and organisms. Modern single-cell atlases routinely represent cells as points embedded in a latent space and then use this space to describe trajectories, cell-state continua, and discrete phenotypic basins that are shared or contrasted across datasets, technologies, and conditions (Haghverdi et al., 2016). Across large consortia and integrative efforts, such latent cell-state landscapes are increasingly treated as the primary substrate for downstream biological interpretation, from developmental lineages and immune activation states to pathological remodeling in disease (Dann et al., 2023).

A key conceptual and empirical observation is that the transcriptomic landscape of single cells is often effectively low-dimensional. Gene expression is structured by co-expression modules, shared regulatory programs, pathway co-activation, and constraints imposed by cell-type identity and tissue context (Crow & Gillis, 2018). This motivates factor-analytic views of single-cell data, in which the high-dimensional gene expression vector for each cell is represented by a relatively small number of latent factors (Lopez et al., 2018). Each factor corresponds to a dimension in a reduced space and is typically interpreted as capturing a coordinated pattern of gene variation, such as a signaling pathway, metabolic program, or differentiation axis. However, factor analysis in the single-cell setting faces a series of challenges: non-Gaussian sampling noise, overdispersion, sparsity, compositional constraints, and the need for both identifiability and biological interpretability of factors (Risso et al., 2018; Kumar et al., 2018; Svensson et al., 2020; Kunes et al., 2024).

For single-cell applications, it is not enough to merely compress data; preserving the geometry of the original space in the low-dimensional representation is equally crucial (Moon et al., 2019). At the global scale, geometric relationships

---

[*]Equal contribution [†]Corr. author. [1]Global Institute of Future Technology, Shanghai Jiao Tong University, Shanghai, China [2]Global College, Shanghai Jiao Tong University, Shanghai, China [3]Gurdon Institute, University of Cambridge, Cambridge, UK [4]Department of Applied Mathematics and Theoretical Physics, University of Cambridge, Cambridge, UK [5]School of Automation and Intelligent Sensing, Shanghai Jiao Tong University, Shanghai, China. Correspondence to: Qiuyu Lian <ql333@cam.ac.uk>, Hongyi Xin <hongyi.xin@sjtu.edu.cn>.

*Proceedings of the 43$^{rd}$ International Conference on Machine Learning*, Seoul, South Korea. PMLR 306, 2026. Copyright 2026

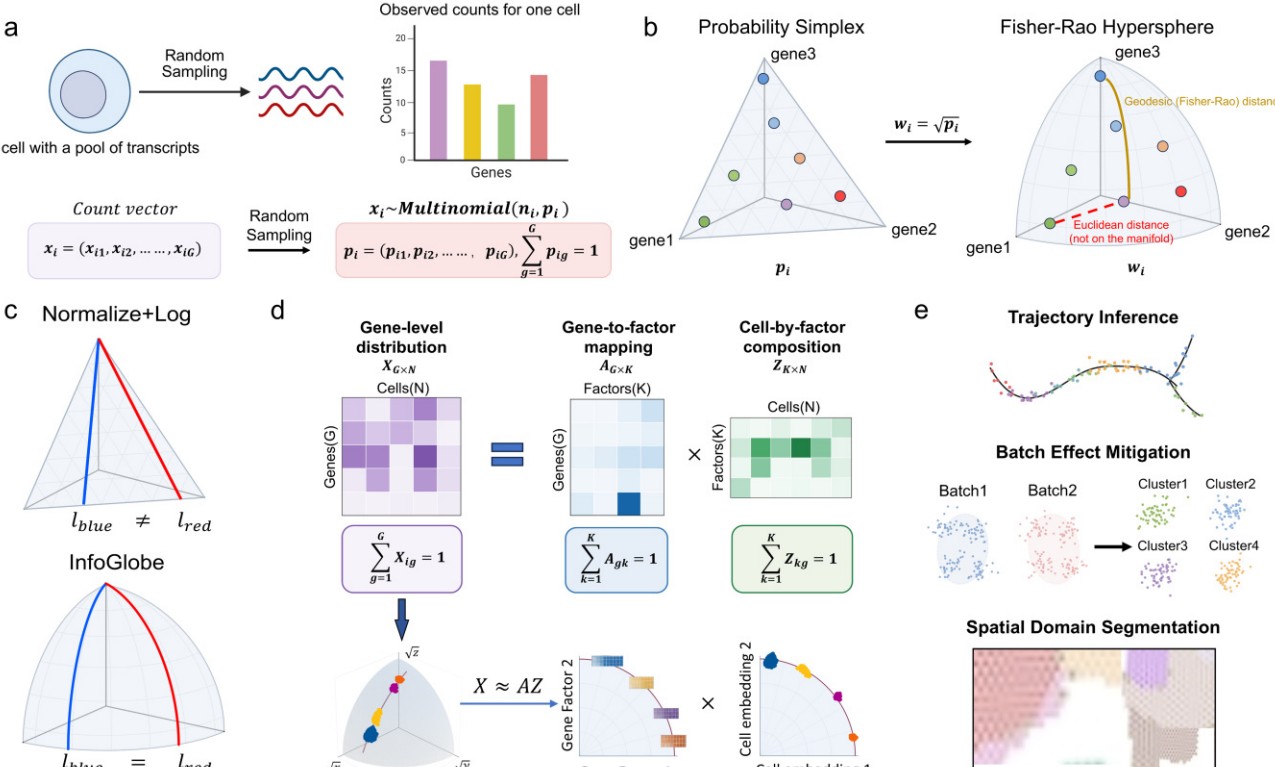

*Figure 1.* Overview of InfoGlobe. (a) Single-cell RNA-seq counts are modeled as samples from a multinomial distribution over genes. (b) The square-root transformation maps the probability simplex to the Fisher-Rao hypersphere, where Fisher-Rao distance is the geodesic distance. (c) Chentsov's theorem identifies the Fisher-Rao metric as the unique invariant Riemannian metric under sufficient statistic-preserving transformations. (d) InfoGlobe uses Fisher-Rao geometry to factorize the cell-gene matrix into gene-to-factor and cell-by-factor matrices, both constrained on multinomial simplices. (e) The resulting factors and embeddings support downstream tasks such as trajectory inference, batch-effect mitigation, and spatial domain segmentation.

between cell populations encode similarity between cell types, lineages, and disease states, and they support tasks such as atlas alignment and cross-condition comparison. At the local scale, geometry determines how subtle within-type heterogeneity, transitional states, and rare subpopulations are organized and separated. When the relationship between latent factors and genes is transparent, so that each latent coordinate can be traced back to coherent gene sets, one can jointly inspect (i) how cells are arranged relative to one another and (ii) which genes or pathways drive these arrangements.

Despite the centrality of geometry, current methods struggle to faithfully capture global relationships between cell types (Kobak & Linderman, 2021). In the ambient high-dimensional gene space, cells are often hypothesized to lie on a nonlinear manifold whose structure is only partially accessible. Conceptually, the biologically meaningful long-range distances between two cell populations should correspond to geodesic distances along this manifold rather than straight-line Euclidean distances in the ambient space (Tenenbaum et al., 2000). In practice, however, the geodesic

metric is unknown. Manifold learning methods therefore approximate it by stitching together local Euclidean neighborhoods, leading to embeddings that preserve local relationships but often distort global structure. Linear methods such as PCA or matrix factorization do provide global distances in the latent space, but these distances do not approximate geodesics on the underlying nonlinear manifold and thus can misrepresent large-scale biological organization.

Here we propose a different perspective to dimension reduction in scRNA-seq, grounded in statistical geometry. Single-cell RNA sequencing can be modeled as stochastic sampling from cell-specific Multinomial distributions over genes. Normalization of gene count data produces a natural statistical manifold defined over the Multinomial parameter space. On this manifold, there exists a canonically defined information-geometric metric, the Fisher-Rao metric(Rao, 1945), that maps the Multinomial distributions of cells to the first-quadrant surface of a unit hypersphere, where geodesic distances between two cells is simply the angular distance on the hypersphere. This information geometry provides a principled notion of long-range similarity

between cells. Moreover, by exploiting the nested closure properties of Multinomial distributions, we show that coarse-graining genes into functional gene groups yields a lower-dimensional Multinomial manifold that is isometric to a corresponding submanifold of the full gene space under the Fisher-Rao metric. Building on these insights, we develop InfoGlobe, an information-geometry-based dimension reduction method that uses high-dimensional Fisher-Rao geodesic distances as supervision. InfoGlobe simultaneously constructs a low-dimensional, geometry-preserving embedding of cells and uncovers interpretable relationships between genes and gene groups, providing a unified framework for global structure preservation, interpretable factorization, and statistically grounded comparisons of cell states.

## 2. Related Works

**Variational autoencoder-based Dimension Reduction models** Several recent methods adopt variational autoencoders (VAEs) to explicitly model the stochastic nature of single-cell count data. scVI assumes a Gaussian latent space with an isotropic prior, which provides a convenient Euclidean representation but does not explicitly account for the geometry of compositional probability distributions. scPhere(Ding & Regev, 2021) replaces the Gaussian prior with a hyperspherical latent distribution to better capture directional structure but the uniform spherical prior enforces isotropic density on the sphere, which can distort intrinsic distance relationships between cells.

More recent approaches such as FlatVI(Palma et al., 2025) and IRVAE(Lee et al., 2022) incorporate pullback metrics induced by the decoder to better respect local geometry in the observation space. These methods introduce decoder-induced pullback metrics to better respect local geometry. However, the resulting geometry is defined implicitly by the generative mapping rather than explicitly grounded in the statistical structure of the data distribution.

**Variance-Driven Euclidean Factorization** Matrix factorization methods are widely used in single-cell analysis to extract interpretable gene programs and functional modules. Both NMF(Lee & Seung, 1999) and GBCD(Liu et al., 2025) operate in Euclidean space and optimize reconstruction-based objectives (e.g., Frobenius loss), which implicitly align components with directions of dominant variance. Consequently, the learned factors are closely tied to the empirical covariance structure of the data and has difficulty in grasping the qualitative and quantitative differences equally well.

## 3. Premises for information-preserving dimension reduction

### 3.1. scRNA-seq as a stochastic sampling process

Existing scRNA-seq protocols profile the transcriptome by sub-sampling a small fraction of mRNA molecules from individual cells. The scRNA-seq process is thus inherently stochastic: the observed count vector for cell $i$, denoted as $\mathbf{x}_i = (x_{i1}, x_{i2}, \ldots, x_{iG}), \sum_g^G x_{ig} = n_i$ is a finite sample drawn from its transcriptome repertoire. This sampling without replacement can be modeled as a hypergeometric sampling process. When total transcriptome $N_i$ is sufficiently large and the sampling fraction is small (i.e., $n_i \ll N_i$), the distribution can be well approximated by a **Multinomial distribution** $\mathbf{x}_i \sim \mathrm{Multi}(n_i, \mathbf{p}_i)$, with the probability density function as follows:

$$P(\mathbf{x}_i \mid \mathbf{p}_i, n_i) = \frac{n_i!}{\prod_{g=1}^{G} x_g!} \prod_{g=1}^{G} p_g^{x_g} \tag{1}$$

Under this formulation, each cell corresponds to a probability vector $\mathbf{p}_i$ lying on the parameter simplex:

$$\Delta^{G-1} = \left\{ \mathbf{p} \in \mathbb{R}^G \,\middle|\, p_g \geq 0, \sum_{g=1}^{G} p_g = 1 \right\}, \tag{2}$$

where $p_g$ denotes the relative abundance of a gene in the transcriptome. Library-size normalization by scaling raw counts to to a constant total, which projects cells onto the surface of this parameter simplex, is mathematically equivalent to maximum likelihood estimation of the Multinomial parameters.

Alternatively, we can also model the mRNA sequencing process as a nested Multinomial sampling process, where transcripts are first sampled by functional gene programs (or factor) $\{f_1, f_2, \cdots f_F\}$ $(F \ll G)$, and then, conditional on each factor, allocated to individual genes according to a secondary, factor-specific Multinomial distribution. By the closure property of Multinomial distributions, the aggregated counts obtained from this two-stage sampling remain Multinomial; when sufficient information about the intermediate allocations is retained, the nested and single-stage formulations are statistically equivalent.

Under this nested formulation, the transcript count of gene $g$ in cell $i$ is given by $x_g = \sum_j f_{j,g}$, where the factor-$j$-specific contribution $f_{j,g} \sim \mathrm{Multi}(f_j, \mathbf{q}_j)$, and the factor-wise transcript counts $\mathbf{f} \sim \mathrm{Multi}(n_i, \mathbf{z}_i)$. Here, $\mathbf{z}_i$ represents the factor expression proportions for cell $i$, while $\mathbf{q}_j$ encodes the gene distribution associated with factor $j$.

From a information-preservation perspective, dimension reduction seeks to **recover these low-dimensional factor**

**proportions** $\mathbf{z}_i$ for each cell $i$, effectively projecting the transcriptomic landscape from the high-dimensional by-gene space to the low-dimensional by-factor space. From a factor analysis perspective, the set $\{\mathbf{q}_1, \mathbf{q}_2, \cdots\}$ denotes the association of genes to each factor, quantifying gene contributions to individual factors and enabling biological interpretation of the underlying gene programs.

### 3.2. Fisher-Rao distance establishes an information geometry for scRNA-seq

The Fisher-Rao distance provides a natural metric for comparing probability distributions, arising intrinsically from the Fisher information matrix interpreted as the metric tensor on the statistical manifold of model parameters. This perspective, which is the cornerstone of information geometry, endows the parameter space with a Riemannian structure where infinitesimal distances encode statistical distinguishability.

A metric tensor is a smooth field of symmetric positive-definite bilinear forms on each tangent space of a manifold, defining lengths $ds^2 = g_{ij}dx^i dx^j$, angles, and geodesics on curved spaces. The Fisher matrix inherits these properties (symmetric, positive semi-definite under regularity conditions), establishing a principled geometry over probability distributions that quantifies information loss between nearby parameter values.

For the Multinomial family, the Fisher information matrix coincides with the metric tensor induced by the square-root pullback embedding of the probability simplex $\Delta^{G-1}$ onto the unit hypersphere. Specifically, the Fisher information matrix takes the form:

$$[I(\mathbf{p})]_{i,j} = N \sum_{k=1}^{G} \frac{1}{p_k} \frac{\partial p_k}{\partial p_i} \frac{\partial p_k}{\partial p_j}. \tag{3}$$

The pullback metric tensor under the square-root map $\mathbf{p} \mapsto 2\sqrt{\mathbf{p}}$ is structurally identical:

$$[\mathbf{M}]_{i,j} = \sum_{k=1}^{G} \frac{1}{4\theta_k} \frac{\partial \theta_k}{\partial e_i} \frac{\partial \theta_k}{\partial e_j}, \tag{4}$$

where $\boldsymbol{\theta} = \mathbf{p}$ parameterizes the simplex coordinates and $\mathbf{e}$ are local tangent-space coordinates on the square-root transformed hypersphere. This equivalence implies that Fisher-Rao distances between Multinomial distributions reduce to **great-circle distances** on the hypersphere after square-root transformation:

$$D_{\text{FR}}(\boldsymbol{p}_1, \boldsymbol{p}_2) = \arccos(\sqrt{\boldsymbol{p}}_1 \cdot \sqrt{\boldsymbol{p}}_2). \tag{5}$$

Additional details on this could be found in the Appendix B.

Beyond providing a principled framework for long-distance measurement, the Fisher-Rao information geometry and associated square-root transformation offer several additional benefits. First, unlike Euclidean distances on a normalized simplex or distances after log transformation, the square-root transform **balances qualitative and quantitative variation**. Log transformation, a widespread preprocessing step in scRNA-seq, is motivated by an emphasis on fold-change-like effects in biological functions, but it disproportionately inflates variation among lowly expressed genes while compressing variation among highly expressed genes. In contrast, the square-root transform gently amplifies variation in low-expression genes without overly suppressing differences among highly expressed genes, yielding a more calibrated representation of transcriptomic changes.

Second, projecting gene expression onto a hypersphere rather than the probability simplex naturally **mitigates spurious correlations arising from the simplex constraint** $\sum p_g = 1$. On the simplex, total-count normalization induces mathematical interdependence between coordinates: an increase in the relative abundance of one gene necessarily decreases others, creating superficial negative correlations under Euclidean geometry. The square-root embedding lifts this degeneracy, yielding geometrically faithful gene-gene relationships. For instance, on a 3-dimensional Reimannian spherical geometry, changes in the expression of a single gene $g_2$ trace a latitude on the sphere, which is spherically orthogonal to proportional shifts in the ratio $g_1 : g_3$. Latitudes (changes in $g_2$) are everywhere perpendicular to longitudes (changes in $g_1 : g_3$), disentangling individual gene variations from proportional rescaling. This geometric independence is critical for interpretable factor analysis, as explored subsequently.

Third, square-root transformation **provides superior variance stabilization for Multinomial counts** compared to log transformation, avoiding geometric distortions at low-expression values. Variance stabilization is a primary goal of scRNA-seq preprocessing, aiming to break the mean-variance dependence inherent in count data. Log transformation works well for Poisson-like distributions where standard deviation SD $\propto$ mean, but Multinomial counts have $\mathbb{E}[X_i] = np_i$ and $\text{Var}(X_i) = np_i(1 - p_i)$, making Var $\propto$ mean. For this dependence structure, square-root transformation is theoretically more optimal. By the delta method, $\text{Var}[\log(p_i)] \approx \frac{1-\pi_i}{n\pi_i^2}$ where $\pi_i = np_i$, which explodes as $\pi_i \to 0$. In contrast, $\text{Var}[\sqrt{p_i}] \approx \frac{1-\pi_i}{4n}$, remaining stably bounded across the expression spectrum and preventing undue inflation of noise from lowly expressed genes.

### 3.3. Statistical manifold extraction as a isometric dimension reduction problem

By the nested closure property of Multinomial distributions, scRNA-seq dimension reduction, if sufficient information is retained, can be viewed as a mapping between two statistical manifolds: a low-dimensional hypersphere

where cells are represented by factor proportions (gene programs); and a high-dimensional hypersphere where cells are parametrized by gene proportions. These manifolds are connected through a set of factor-to-gene multinomial distributions $A = \{\mathbf{q}_j\}$, assumed constant across cells[1].

Under the strict non-overlapping assumption (each gene belongs to exactly one factor), the low-dimensional by-factor manifold is geometrically isometric to its high-dimensional by-gene counterpart in every tangent space. This follows directly from Chentsov's theorem, which states that sufficient-statistics-preserving maps between exponential families induce isometric embeddings of their Fisher-Rao geometries. Conceptually, with fixed non-overlapping gene-to-factor assignments, variations in the high-dimensional (gene) manifold are simply replications of changes in the low-dimensional (factor) manifold. The arch (angular) distance between two points, $\mathbf{z}_1$ and $\mathbf{z}_2$ in the by-factor space, and their projected images $\mathbf{x}_1$ and $\mathbf{x}_2$ in the high dimensional by-gene space, are linearly proportional by a factor $\sqrt{|\mathbf{q_j}|}$ (number of genes participating in the factor). Geometrically, the factor manifold embeds into the gene manifold via a Markov embedding $A$, which lifts cell-by-factor coordinates to cell-by-gene coordinates through the fixed factor-to-gene mapping, preserving local Fisher-Rao geometry. **The goal of dimension reduction** is thus to identify **both the Markov embedding $A$** and **the low-dimensional cell-by-factor Multinomial parameters $\mathbf{z}_i$** for every cell $i$, such that the low-dimensional hypersphere remains geometrically isometric to the high-dimensional hypersphere in every tangent space.

## 4. The InfoGlobe Model

### 4.1. Information-preserving factorization

Motivated by Chentsov's Theorem, we propose a Fisher–Rao geometry-aware matrix factorization framework. Specifically, Given an observed gene counts matrix $X$ (after $L1$ normalization), our goal is to learn $A$ and $Z$ such that

$$X = AZ, \tag{6}$$

where $X$ is the high-dimensional observation, $A$ denotes the gene-to-factor relation matrix and $Z$ is the latent embedding of cells. Each column of $X, Y, Z$ represents a parameter vector of a multinomial distribution (summing to one).

### 4.2. Geometry Preserved Loss

First, we enforce a Fisher-Rao distance based reconstruction loss that aligns each observed gene distribution with its

reconstructed counterpart,

$$L_{\text{rec}} = \sum_i (d_{\text{FR}}(x_{\cdot i}, (AZ)_{\cdot i})) \tag{7}$$

Here, $d_{FR}(x_{\cdot i}, (AZ)_{\cdot i})$ denotes the Fisher-Rao distance between $x_{\cdot i}$ and $(AZ)_{\cdot i}$, which serve as parameters for the multinomial distribution.

Second, to preserve the global structure of cells on the manifold, we match pairwise Fisher–Rao distances between the original space and the low-dimensional functional space in a MDS manner,

$$L_{\text{geom}} = \sum_{i,j} (d_{\text{FR}}(x_{\cdot i}, x_{\cdot j}) - d_{\text{FR}}(z_{\cdot i}, z_{\cdot j}))^2 \tag{8}$$

encouraging the embedding to retain intrinsic cell-cell relationships. Although the Fisher-Rao distance does not admit closed-form expressions for many distribution families, it is analytically tractable for Multinomial distributions(Appendix C).

The final objective combines these two terms, $L = L_{rec} + \lambda \times L_{geom}$, with a Lagrangian multiplier $\lambda$, yielding a geometry-aware matrix decomposition that simultaneously reconstructs gene expression and preserves the global manifold structure of the data.

### 4.3. Geometry-preservation testing for Informative factor identification

Unlike PCA, the factors learned by InfoGlobe are not ordered by explained variance, and increasing the factor count may refine an existing factor into several highly correlated sub-factors rather than reveal genuinely new biological degrees of freedom. We therefore select the number of factors by jointly monitoring geometric fidelity and factor redundancy. Specifically, for each candidate factor count $K$, we train InfoGlobe and record

1. the Riemannian MDS loss, measuring the discrepancy between pairwise Fisher-Rao distances in the original and learned manifolds; and

2. the maximum Pearson correlation among cell-factor loadings, measuring redundancy induced by factor splitting.

When $K$ is too small, the latent manifold lacks sufficient flexibility and the MDS loss remains high. As $K$ approaches the intrinsic dimensionality of the data, the MDS loss decreases and then saturates. Further increasing $K$ typically yields redundant sub-factors with strongly correlated cell loadings while providing little additional improvement in geometric preservation. We therefore choose the smallest $K$ at which the MDS loss has stabilized before a marked

---

[1]This shared factor-to-gene structure is a modeling assumption of InfoGlobe, elaborated later in Discussion.

increase in inter-factor correlation, favoring the most parsimonious representation that preserves the geometry of the data, as shown in Figure 8).

### 4.4. Correlation testing for biological factors, condition factors and batch factors deconvolution

InfoGlobe represents each cell by $K$ nonnegative factors that sum to one, where $\tilde{z}_{fi}$ can be interpreted as the proportion of the cell $i$ assigned to functional program $f$. In multi-batch data, technical variation typically concentrates on a small subset of factors: these factors show pronounced shifts across batches, yet provide little discrimination between biological cell types. We leverage this sparsity to integrate batches by detecting batch-associated factors and removing only their batch-driven mean shifts, while leaving the remaining factors unchanged.

Let $b_i \in \{1, \ldots, B\}$ denote the batch label. For each factor $f \in \{1, \ldots, K\}$, we test whether its mean differs across batches using a one-way ANOVA, i.e., $H_0 : \mu_{f,1} = \cdots = \mu_{f,B}$ with $\mu_{f,b} = \mathbb{E}[z_{if} \mid b_i = b]$, and mark $f$ as batch-associated if the resulting $p$-value is below a preset level $\alpha$. For each marked factor, we residualize batch effects via a batch-indicator regression

$$z_{if} = \beta_{0f} + \sum_{b=2}^{B} \beta_{bf} \, \mathbb{I}(b_i = b) + \varepsilon_{if}, \qquad (9)$$

and replace $z_{if}$ by the residual

$$z'_{if} = z_{if} - \widehat{\mathbb{E}}[z_{if} \mid b_i]. \qquad (10)$$

Unmarked factors are left unchanged. Finally, we re-normalize $\mathbf{z}'_i$ back to the simplex,

$$\hat{\mathbf{z}}_i = \frac{\max(\mathbf{z}'_i, 0)}{\|\max(\mathbf{z}'_i, 0)\|_1} \in \Delta^{K-1}, \qquad (11)$$

and use $\hat{\mathbf{z}}_i$ for downstream visualization, distance computation, and analysis.

## 5. Results

### 5.1. Information-preserving dimension reduction

We validate the information-preservation performance of InfoGlobe through simulation following the nested Multinomial distribution model. In this simulation, each cell is assigned a simplex-valued factor-usage composition $q_i \in \Delta^{k-1}$. A fixed gene-to-factor relation matrix $A$ lifts these latent compositions to gene-level probabilities via $p_i = Aq_i$. Observed counts are then randomly sampled following the Multinomial distribution governed by $p_i$, producing noisy high-dimensional profiles. The low-dimensional cell-by-factor matrix serves as the ground truth.

*Table 1.* Local and global structure preservation on simulated data, measured by Trustworthiness, Continuity, and Spearman correlation.

| | TRUST.($\uparrow$) | CONT.($\uparrow$) | SPEARMAN.($\uparrow$) |
|---|---|---|---|
| SCPHERE | $0.69_{\pm 0.01}$ | $0.68_{\pm 0.01}$ | $0.69_{\pm 0.01}$ |
| SCVI | $0.74_{\pm 0.01}$ | $0.69_{\pm 0.01}$ | $0.36_{\pm 0.01}$ |
| FLATVI | $0.75_{\pm 0.00}$ | $0.71_{\pm 0.01}$ | $0.45_{\pm 0.01}$ |
| IRVAE | $0.73_{\pm 0.00}$ | $0.69_{\pm 0.01}$ | $0.51_{\pm 0.01}$ |
| INFOGLOBE | $\mathbf{0.83}_{\pm 0.00}$ | $\mathbf{0.84}_{\pm 0.01}$ | $\mathbf{0.88}_{\pm 0.01}$ |

*Table 2.* Clustering performance of different embeddings measured by ARI, V-measure, and FMI.

| | ARI.($\uparrow$) | V-MEASURE.($\uparrow$) | FMI.($\uparrow$) |
|---|---|---|---|
| SCPHERE | $0.50_{\pm 0.09}$ | $0.65_{\pm 0.01}$ | $0.76_{\pm 0.02}$ |
| SCVI | $0.55_{\pm 0.03}$ | $0.67_{\pm 0.02}$ | $0.80_{\pm 0.09}$ |
| FLATVI | $0.58_{\pm 0.03}$ | $0.64_{\pm 0.03}$ | $0.77_{\pm 0.02}$ |
| IRVAE | $0.53_{\pm 0.01}$ | $\mathbf{0.69}_{\pm 0.03}$ | $0.80_{\pm 0.03}$ |
| INFOGLOBE | $\mathbf{0.70}_{\pm 0.02}$ | $0.69_{\pm 0.02}$ | $\mathbf{0.84}_{\pm 0.01}$ |

InfoGlobe is compared against scPHERE, scVI, FlatVI, and IRVAE. We evaluate local (Trustworthiness, Continuity), global (Spearman $\rho$) distance preservation. All methods faithfully retain local neighborhood relationships, but InfoGlobe shows pronounced improvement in global structure preservation. Overall, InfoGlobe is uniquely positioned to preserve long-range geometric ranking, as evidenced by its significantly improved Spearman score. The improved low-dimensional distance relationships also enhance clustering performance. We tested Leiden clustering performance across all embeddings using: ARI (clustering agreement), V-Measure (homogeneity and completeness), FMI (geometric mean of precision and recall). Results are summarized in Table 2. InfoGlobe embeddings consistently produce the most accurate clustering results.

### 5.2. Harmonization of qualitative and quantitative variations

**InfoGlobe Captures qualitative and quantitative gene programs.**

To validate the factorization performance of InfoGlobe, we generated a synthetic dataset again following a nested Multinomial mixture distribution. In this simulation, factors are categorized as: quantitative factors: expressed across multiple cell types with varying expression strengths; qualitative factors: on/off factors exclusive to single cell types. The dataset comprises 2,400 cells $\times$ 200 genes across five cell types.

InfoGlobe is compared against NMF and GBCD. Results are shown in Figure 2. InfoGlobe consistently and accurately attributes both qualitative (on/off) and quantitative (varying strength) genes to their correct factors while recovering true

expression strengths. In contrast, NMF and GBCD are less stable, particularly struggling to distinguish qualitative from quantitative factors.

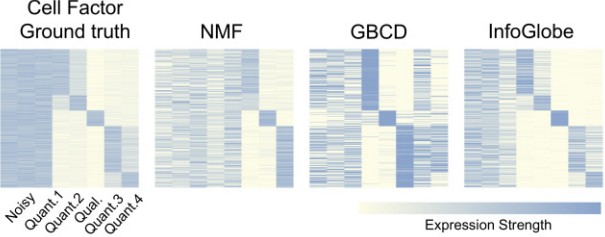

*Figure 2.* **Qualitative versus quantitative gene factor discovery**. NMF and GBCD both identified noise-gene-factor associations but conflated quantitatively distinct factors (Quant.3 and Quant.4) into shared latent dimensions, whereas InfoGlobe recovered all factors distinctly.

**InfoGlobal enables consistent and versatile factorization of gene expression.** To test InfoGlobe's stability, we removed two cell types and their associated factors from the previous simulation, expecting the cell-factor matrix to remain largely unchanged. Results (Figure 3) confirm this: InfoGlobe embeddings show minimal changes, while NMF and GBCD cell-factor embeddings experience profound alterations. This proves the superior stability of the InfoGlobe factorization and increases confidence in its robustness.

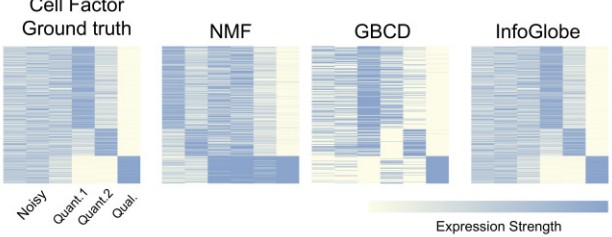

*Figure 3.* **Stability comparison under cell type proportion variation** When cellular composition changes, previously identified factors corresponding to unperturbed cell populations by NMF and GBCD are distorted or lost. In contrast, InfoGlobe, by preserving global geometric structure through Fisher-Rao geodesics, maintains stable factor representations for invariant cell types, demonstrating robustness to compositional variation.

**InfoGlobe Improves downstream trajectory analysis.** We applied InfoGlobe to a public T-cell-depleted bone marrow single-cell dataset(Otto et al., 2024) representing human hematopoietic differentiation. During hematopoietic differentiation, cellular states evolve through gradual transcriptional reprogramming, where coordinated changes in multiple gene programs drive continuous transitions rather than abrupt shifts. Their developmental progression is expected to form continuous trajectories without discontinuities in the latent space. We applied PHATE using two embeddings

as input: Euclidean representation derived from PCA and the statistical manifold learned by InfoGlobe.

The visualization based on InfoGlobe reveals smooth and coherent developmental trajectories, where cells that are temporally distant along differentiation are also well separated in the embedding space, consistent with biological expectations (Figure 4A). In contrast, the PCA-based representation introduces noticeable distortions: the B-cell lineage exhibits fragmented trajectories, and the erythroid lineage shows mixing at late differentiation stages. These artifacts contradict known biological progression and indicate that Euclidean embeddings fail to preserve the global structure of cellular relationships. A comprehensive comparison of the PHATE visualizations for the remaining methods is provided in Figure 9.

We next applied InfoGlobe to investigate transcriptional programs underlying erythroid cell differentiation. By analyzing factor activities along the inferred developmental trajectory, we identified a set of trajectory-associated factors whose activations vary smoothly over the development trajectory and delineate distinct stages of lineage progression (Figure 4B). Several factors exhibited stage-specific dynamics, with early-acting programs enriched in stem and progenitor states and late-acting programs progressively activated in committed erythroid-cell populations. The complete results for the three trajectory analyses are shown in Figure 10. Notably, these factors include known regulators of erythroid-cell specification and maturation(Figure 4C), indicating that the learned representations capture biologically meaningful transcriptional programs rather than dataset-specific variance. The distributions of the corresponding genes on the UMAP embeddings are shown in Figure 11 and Figure 12.

### 5.3. Mitigation of sequencing-depth-variation-induced batch effects

To evaluate performance on multi-batch datasets, we validated InfoGlobe on the pancreas dataset(Luecken et al., 2022), which comprises nearly ten batches and dozens of cell types, presenting a substantial challenge for effective batch integration.

Benefiting from its factorized probabilistic formulation, InfoGlobe provides a more interpretable alternative. During matrix decomposition, InfoGlobe automatically separates variation into distinct functional programs, allowing both cell-type–specific and batch-specific factors to emerge(Figure 5 B, Figure 13). By inspecting the genes with high loadings in each factor, batch-associated signals can be explicitly identified and attributed to concrete gene sets. This enables transparent and biologically interpretable batch integration, while preserving meaningful cellular structure(Figure 5 A).

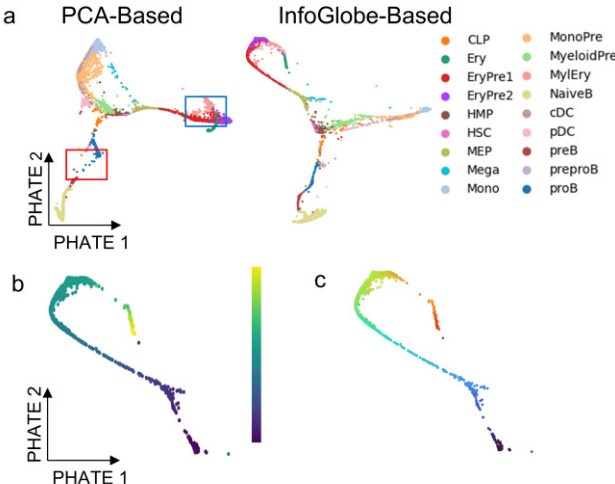

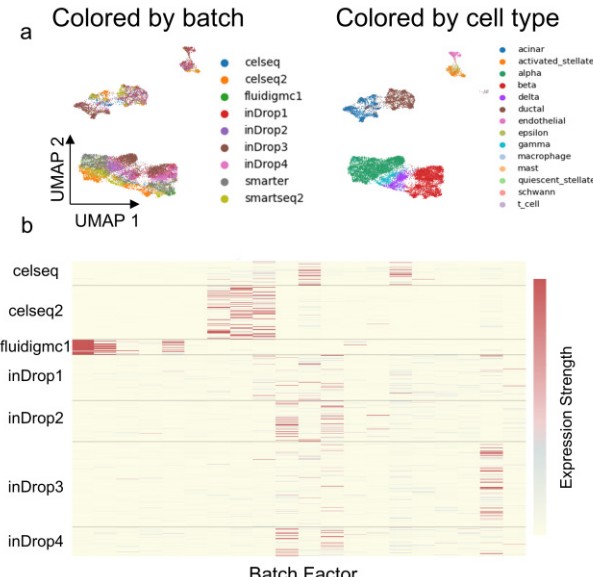

*Figure 4.* (a) PHATE embeddings computed from (left) PCA-derived representation and (right) InfoGlobe-derived representation. (b) Pseudo time trajectory for erythroid lineage within Palantir. Color scale indicates progression along the differentiation trajectory. (c) Erythroid lineage differentiation factor identified by InfoGlobe. Cell color is computed as a weighted combination of multiple time factors, where weights reflect each cell's loading across temporal factors. The coloring aligns with independent pseudo time analysis, validating the biological relevance of identified time factors.

*Figure 5.* (a) UMAP embeddings of raw (left) and batch-corrected integrated (right) data. Left: cells colored by batch identity; Right: cells colored by annotated cell type. (b) Batch-specific factors identified by InfoGlobe across diverse single-cell datasets. Color intensity indicates factor activation strength.

## 5.4. Spatial domain segmentation enhancement via quantitative variation amplification

Finally, we evaluated InfoGlobe's compatibility with spatial transcriptomics dataset(Birk et al., 2025a) (ST). Unlike scRNA-seq, ST spots aggregate multiple cell types, resulting in much more attenuated transcriptomic differences. To test whether InfoGlobe improves ST domain segmentation, we compared SpaGCN (Hu et al., 2021), a-convolution-graph-neural-network-based segmentation algorithm, using PCA vs. InfoGlobe embeddings on an annotated breast cancer Visium slice (Birk et al., 2025b).

The results are summarized in Figure 6. In comparison against PCA embeddings, SpaGCN domain segmentation with InfoGlobe is significantly better aligned with the annotation ground truth than PCA embeddings. The complete embedding results for all methods are shown in Figure 14. Notably, SpaGCN with PCA embeddings cannot correctly identify the DCIS/LCIS versus IDC boundry (blue box). This is likely due to more nuanced gradual changes in transcriptome as lesion transitions from dormant to invasive. Similarly, Tumor edge versus Healthy boundary (red box) is also not correctly identified, likely due to the more nuanced and qualitative differences between cancerous epithlium and its normal counterpart. Replacing the PCA embedding with InfoGlobe yields a consistent gain in segmentation (ARI: $0.48 \rightarrow 0.56$, NMI/AMI increasing accordingly Table 3). Visually, InfoGlobe embeddings also helps delineating the

two previously-confounded tumor boundaries, in addition to cleaner separation of domains in the lower-left and upper-right regions. Also with InfoGlobe features, SpaGCN attains the best ARI in this benchmark while remaining competitive on NMI/AMI against BayesSpace and consistently outperforming SEDR (Table 3) (Zhao et al., 2021; Xu et al., 2024).

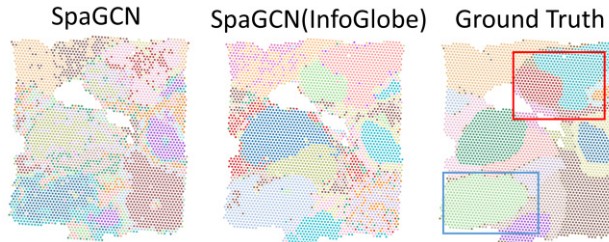

*Figure 6.* **Spatial domain segmentation results** over a breast cancer Visium spatial transcriptomic slice. From left to right: SpaGCN, SpaGCN with InfoGlobe embeddings as input, and pathologist-guided ground truth. In the ground-truth panel: Blue boxes highlight the two major confusion pairs (DCIS/LCIS vs. IDC), and red boxes highlight Tumor edge vs. Healthy.

## 6. Conclusion and Discussion

In this study, we present InfoGlobe, a statistical dimension reduction framework that maps high-dimensional cell expression observations to a low-dimensional gene program factor space. InfoGlobe approaches the dimension reduction problem from a information geometry perspective, where it

*Table 3.* Clustering performance of different embeddings measured by ARI, V-measure, and FMI.

| | ARI.($\uparrow$) | NMI.($\uparrow$) | AMI.($\uparrow$) |
|---|---|---|---|
| BAYESSPACE | 0.51 | **0.63** | **0.62** |
| SEDR | 0.37 | 0.55 | 0.54 |
| SPAGCN | 0.48 | 0.58 | 0.57 |
| SPAGCN(INFOGLOBE) | **0.56** | 0.62 | 0.61 |

maps high-dimensional multinomial distributions to a low-dimensional hypersphere while maximally preserving the information geometry. By an extending the Chentsov's theorem, the Markov embedding can be perceived as a factor-to-gene mapping, establishing the contribution of genes in co-expression gene programs. Experiments shows that InfoGlobe correctly characterizes both local and global geometry; harmonizes quantitative and qualitative variations; and accurately attribute genes to gene programs. These results prove that InfoGlobe provides a unified framework for global structure preservation, interpretable factorization, and statistically grounded comparisons of cell states. A key limitation of InfoGlobe is its assumption that gene-to-factor mapping is globally constant. While this assumption is not unique to InfoGlobe, it is a strong assumption that could limit its applicability. We seek to extend the flexibility of InfoGlobe in our future work.

**InfoGlobe** is available at https://github.com/gift-novellab/InfoGlobe.

## Acknowledgements

This work is supported by STI2030-Major Projects 2022ZD0212400, Lingang Laboratory grant LGL-8888, STCSM grant 24510714300 and 20DZ2254400, Guang-Dong Basic and Applied Basic Research Foundation grant 2023B1515120006 and SJTU Science-Medicine interdisciplinary grant YG2026ZD09 and 24X010301456. We also thank Prof. Jingyi Jessica Li, Prof. Binzhi Qian and Prof. Hongyu Zhao for helpful discussions and insightful suggestions.

## Impact Statement

InfoGlobe provides a geometry-faithful embedding of scRNA-seq data by grounding cell-cell distances in information geometry, yielding a low-dimensional representation that remains consistent with the underlying probabilistic generative structure. This embedding serves as a reliable backbone for downstream analyses that rely on both local neighborhoods and long-range relationships.

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

## A. Code and Datasets

We have made our code publicly available at https://github.com/gift-novellab/InfoGlobe. All datasets used in this study are open source, and their associated publications are cited in the manuscript.

## B. Geodesic Distance and Fisher–Rao Metric

Quantifying distances between cells in single-cell transcriptomics requires a metric that respects the underlying geometry of gene expression distributions. Riemannian geometry, and in particular the Fisher-Rao metric from information geometry, offers a principled approach by defining geodesic-aware distances directly on the statistical manifold of a family of parameterized distributions.

A **Riemannian metric** tensor $g_p$ assigns a positive-definite inner product on the tangent space $T_p\mathcal{M}$ of a smooth manifold $\mathcal{M}$ at a coordinate $p$ in an Euclidean space:

$$g_p : T_p\mathcal{M} \times T_p\mathcal{M} \to \mathbb{R}. \tag{12}$$

The real Euclidean length of a tangent vector $v \in T_p\mathcal{M}$ on the Riemannian manifold, defined as the inner product of the vector, can be computed as:

$$\|v\|_p = \sqrt{g_p(v,v)}, \tag{13}$$

and the geodesic distance between two points $p, q \in \mathcal{M}$ on the non-Euclidean smooth manifold is the length of the shortest geodesic curve connecting them:

$$d(p,q) = \inf_{\gamma \,:\, \gamma(0)=p,\gamma(1)=q} \int_0^1 \sqrt{g_{\gamma(t)}(\dot{\gamma}(t),\dot{\gamma}(t))}\, dt. \tag{14}$$

For parametrized distributions, the Fisher information matrix serves as a natural metric tensor that defines a Riemannian geometry in the parameter space. Intuitively, Fisher information matrix, defined as:

$$[I(\theta)]_{i,j} = \mathbb{E}_{x\sim p(x;\theta)}\left[\left(\frac{\partial}{\partial\theta_i}\log p(x;\theta)\right)\left(\frac{\partial}{\partial\theta_j}\log p(x;\theta)\right)\Big|\theta\right] \tag{15}$$

Given the natural equivalence of the Fisher information matrix and the Riemannian metric tensor, we can harness the Fisher information matrix as defining a mapping between a Riemannian geometry, in which the distributions locates on a Riemannian surface, and the Euclidean parameter space. The distance, known as the **Fisher-Rao distance**, between two distributions $p(x;\theta_1)$ and $p(x;\theta_2)$ with respective parameters can thus be computed as the geodesic distance of the shortest path on the Riemannian statistical manifold:

$$D_{\mathrm{FR}}(\theta_1,\theta_2) = \inf_{\gamma} \int_0^1 \sqrt{\dot{\gamma}(t)^\top I(\gamma(t))\dot{\gamma}(t)}\, dt, \tag{16}$$

where $\gamma(t)$ is a smooth path connecting $\theta_1$ and $\theta_2$.

## C. Fisher Information Metric for the Multinomial Distribution

Let $\theta$ denote the $n$-dimensional Cartesian coordinates of points on a simplex, where $|\theta|_1 = 1$. Let $\omega$ denote the square-root transformed Cartesian coordinates, in which the simplex is transformed into a sphere with $|\omega|_2 = 1$. For a point on the sphere, let the bases $\mathbf{e_1}, \mathbf{e_2}, \ldots, \mathbf{e_n}$ denote the orthogonal bases of the local tangent space. A vector $\mathbf{z}$ in the local tangent space, represented as $\mathbf{z} = \sum_{i=1}^n \beta_i \mathbf{e_i}$, can be converted to a representation in the $\omega$ space as $\mathbf{z} = \sum_{i=1}^n \alpha_i \omega_\mathbf{i}$ with chain rule, where $\alpha_i = \sum_{i=1}^n \beta_i \frac{\partial \omega_\mathbf{i}}{\partial \mathbf{e_i}}$. The inner product of a vector $\mathbf{z}$ in the $\omega$ coordinate system can thus be translated to the local tangent space as following:

$$|\mathbf{z}|_2 = \sum_{i=1}^n \sum_{j=1}^n \sum_{k=1}^n \beta_i\beta_j \frac{\partial\omega_\mathbf{k}}{\partial\mathbf{e_i}} \frac{\partial\omega_\mathbf{k}}{\partial\mathbf{e_j}} \tag{17}$$

The above equation can be perceived as a translation of L-2 norm between the local tangent space and the $\omega$ coordinate system, as $|\mathbf{z}|_2 = \mathbf{z}^\top \mathbf{G}\mathbf{z}$, where $\mathbf{G}$ is the metric tensor defined as $G_{i,j} = \sum_{k=1}^{n} \frac{\partial \omega_\mathbf{k}}{\partial \mathbf{e_i}} \frac{\partial \omega_\mathbf{k}}{\partial \mathbf{e_j}}$. By substituting in $\omega_i = \sqrt{\theta_i}$ and applying chain rule in reverse ($\theta_i = \omega_i^2$), we have:

$$\mathbf{G}_{i,j} = \sum_{k=1}^{n} \frac{\partial \omega_\mathbf{k}}{\partial \mathbf{e_i}} \frac{\partial \omega_\mathbf{k}}{\partial \mathbf{e_j}} = \sum_{k=1}^{n} \left(\frac{1}{2\omega_\mathbf{k}} \frac{\partial \theta_\mathbf{k}}{\partial \omega_\mathbf{k}} \frac{\partial \omega_\mathbf{k}}{\partial \mathbf{e_i}}\right)\left(\frac{1}{2\omega_\mathbf{k}} \frac{\partial \theta_\mathbf{k}}{\partial \omega_\mathbf{k}} \frac{\partial \omega_\mathbf{k}}{\partial \mathbf{e_j}}\right) = \frac{1}{4\theta_k} \sum_{k=1}^{n} \frac{\partial \theta_\mathbf{k}}{\partial \mathbf{e_i}} \frac{\partial \theta_\mathbf{k}}{\partial \mathbf{e_j}} \tag{18}$$

given that $\frac{\partial \theta_\mathbf{k}}{\partial \mathbf{e_i}} = \frac{\partial \theta_\mathbf{k}}{\partial \omega_\mathbf{k}} \frac{\partial \omega_\mathbf{k}}{\partial \mathbf{e_i}} = 2\omega_\mathbf{k} \frac{\partial \omega_\mathbf{k}}{\partial \mathbf{e_i}}$. Equation 18 establishes the metric tensor between a simplex and its square-root-transformed unit hypersphere, where the geodesic distance on the equivalent hypersphere manifold between two points $\theta_1$ and $\theta_2$ on the hyper-simplex can be computed as:

$$D_{\mathrm{FR}}(\theta_1, \theta_2) = arccos(\omega_1 \cdot \omega_2) \tag{19}$$

with $\theta_1 = (\omega_1)^2$ and $\theta_2 = (\omega_2)^2$.

For Multinomial distributions, the Fisher information matrix takes the same form as the metric tensor for the simplex-to-sphere pullback mapping. To see this, we use an alternative form of the Fisher information matrix

$$[I(\theta)]_{i,j} = -\mathbb{E}_{x \sim p(x;\theta)} \left[ \left( \frac{\partial}{\partial \theta_i \partial \theta_j} \log p(x;\theta) \right) \bigg| \theta \right] \tag{20}$$

By plugging Equation 1 into Equation 20 and apply chain rule, we have:

$$\frac{\partial}{\partial p_i \partial p_j} \log p(x; \mathbf{p}) = \sum_{k=1}^{G} \frac{\partial}{\partial p_i} \left( \frac{\partial p_k}{\partial p_j} \cdot \left( \frac{\partial \log p(x; \mathbf{p})}{\partial p_k} \right) \right) = \sum_{k=1}^{G} \frac{\partial}{\partial p_i} \left( \frac{\partial p_k}{\partial p_j} \cdot \left( \frac{x_k}{p_k} \right) \right)$$
$$= \sum_{k=1}^{G} \frac{\partial}{\partial p_i \partial p_j} \left( p_k \frac{x_k}{p_k} \right) - \sum_{k=1}^{G} \frac{\partial p_k}{\partial p_i} \cdot \frac{\partial p_k}{\partial p_j} \cdot \left( \frac{x_k}{p_k^2} \right) \tag{21}$$

Take 21 into 20, we have:

$$[I(\theta)]_{i,j} = -\sum_{k=1}^{G} \frac{\partial \mathbb{E}[x_k]}{\partial p_i \partial p_j} + \sum_{k=1}^{G} \frac{\partial p_k}{\partial p_i} \cdot \frac{\partial p_k}{\partial p_j} \cdot \left( \frac{\mathbb{E}[x_k]}{p_k^2} \right). \tag{22}$$

Notice that $\mathbb{E}[x_k] = N \cdot p_k$ with $N$ being the total RNA molecule count of a cell. Thus Equation 22 becomes

$$[I(\theta)]_{i,j} = \frac{\partial}{\partial p_i \partial p_j} \sum_{k=1}^{G} (x_k) + \sum_{k=1}^{G} \frac{\partial p_k}{\partial p_i} \cdot \frac{\partial p_k}{\partial p_j} \cdot \left( \frac{N}{p_k} \right) = \frac{N}{p_k} \sum_{k=1}^{G} \frac{\partial p_k}{\partial p_i} \cdot \frac{\partial p_k}{\partial p_j}. \tag{23}$$

Notice that Equation 23 has the same form as Equation 18 but amplified by a factor of $4N$. This establishes the geometric equivalence between information distance and arc distance: The Fisher-Rao distance between two Multinomial distributions is a constant multiple of the arc distance between $\omega_i$ and $\omega_j$ on the hypersphere, after taking square-root transformation: $\omega_i^2 = \mathbf{p}_i$ and $\omega_j^2 = \mathbf{p}_j$.

## D. Chentsov's Theorem and Fisher Metric Invariance

In this section we provide the formal statement of Čencov's theorem and explain its implication for Multinomial models used in our formulation.

**Statistical manifolds.** Let $\mathcal{P} = \{p(x; \theta) : \theta \in \Theta\}$ be a parametric family of probability distributions, where $\Theta$ is a smooth parameter space. The family $\mathcal{P}$ forms a statistical manifold, on which one may define Riemannian metrics to measure infinitesimal distances between distributions.

**Chentsov's Theorem.** *Up to a positive constant scaling, the Fisher information metric is the unique Riemannian metric on a statistical manifold that is invariant under sufficient-statistics transformations (equivalently, under probabilistically meaningful mappings such as aggregation, coarse-graining, or reparameterization of outcomes).*

The theorem implies that if one probability distribution is obtained from another by aggregating or regrouping outcomes without introducing additional information, then the intrinsic geometry — and hence distances measured by the Fisher–Rao metric — remain unchanged. By contrast, Euclidean distances or divergence-based measures generally depend on the chosen representation.

**Theorem for For Multinomial Distributions**. Probability vectors lie on the simplex

$$\Delta^{G-1} = \{\mathbf{p} \in \mathbb{R}^G : p_g \geq 0, \sum_g p_g = 1\}$$

A mapping between Multinomial families is a sufficient-statistics transformation if it corresponds to a stochastic aggregation of categories. Concretely, suppose genes are partitioned into disjoint groups $\{C_k\}_{k=1}^K$ such that

$$\bigcup_{k=1}^K C_k = \{1, \ldots, G\}, \qquad C_k \cap C_{k'} = \varnothing \ (k \neq k').$$

This defines an aggregation map that converts fine-grained gene probabilities into coarse functional probabilities (or vice versa) without mixing information across groups.

Such transformations are precisely the Markov morphisms considered in Chentsov's theorem. Consequently, the Fisher information metric is preserved between the corresponding Multinomial manifolds. Therefore, Fisher–Rao distances remain invariant between the low-dimensional functional simplex and the induced high-dimensional gene-expression simplex.

This invariance justifies the use of Fisher–Rao distances when comparing cells across different probabilistic representations, and underlies the geometry-preserving factorization framework proposed in this work.

## E. Simulation Setting

We simulate scRNA-seq count matrices from a probabilistic generative process with an explicit low-dimensional ground truth, designed so that the global geometry is determined by simplex-valued program compositions.

We begin by drawing cell-type proportions $\boldsymbol{\pi} \sim \mathrm{Dirichlet}(\boldsymbol{\alpha})$ and assigning each cell a type $c_i \sim \mathrm{Categorical}(\boldsymbol{\pi})$; marginally, the resulting type counts follow a Dirichlet–multinomial distribution, capturing over-dispersion in population composition relative to a fixed multinomial mode. Conditioned on $c_i$, each cell is endowed with a program-usage vector $\boldsymbol{q}_i \in \Delta^{k-1}$ drawn from a type-specific Dirichlet prior, $\boldsymbol{q}_i \sim \mathrm{Dirichlet}(\boldsymbol{\gamma}_{c_i})$, which serves as the ground-truth low-dimensional representation.

To generate gene-level probabilities, we use a sparse gene-to-factor relation matrix $A \in \mathbb{R}^{G \times k}$ with nonnegative, column-stochastic structure. For each program $j$, we choose a small support set $S_j \subset \{1, \ldots, G\}$ and sample the nonzero weights $\{A_{gj}\}_{g \in S_j}$ from a Dirichlet distribution, setting $A_{gj} = 0$ for $g \notin S_j$ and enforcing $\sum_{g=1}^G A_{gj} = 1$. This construction yields gene probabilities for cell $i$ as

$$\boldsymbol{p}_i = A \boldsymbol{q}_i \in \Delta^{G-1}. \tag{24}$$

Finally, we draw a per-cell library size $n_i$ from a log-normal distribution and sample observed counts

$$n_i \sim \mathrm{LogNormal}(\mu, \sigma^2), \qquad \boldsymbol{x}_i \sim \mathrm{Multinomial}(n_i, \boldsymbol{p}_i), \tag{25}$$

a widely used abstraction that reflects strong depth variability and yields a deliberately shallow, noise-dominated regime. This design ensures a traceable lift from simplex ground truth to high-dimensional observations, enabling direct assessment of whether an embedding recovers the latent compositions and the geometry induced by the generative mechanism.

## F. Running time benchmark

We test the running time of InfoGlobe with a 5090 GPU, 700 genes. The top-left panel shows the time required for InfoGlobe to converge as the number of sampled cells increases from 200 to 1000. The time increases almost linearly as more cells are added. The subsequent panels show the factor- cell heatmaps for different numbers of sampled cells (500, 1000, 2000, and 3000). These heatmaps illustrate the learned factor-cell matrices for each sampled cell count. The results demonstrate that

as the number of cells increases, the learned factor representations become increasingly consistent, indicating that the results stabilize and are largely unaffected by the number of sampled cells after a certain point. This shows that InfoGlobe scales efficiently without significant loss in quality as the cell number grows.

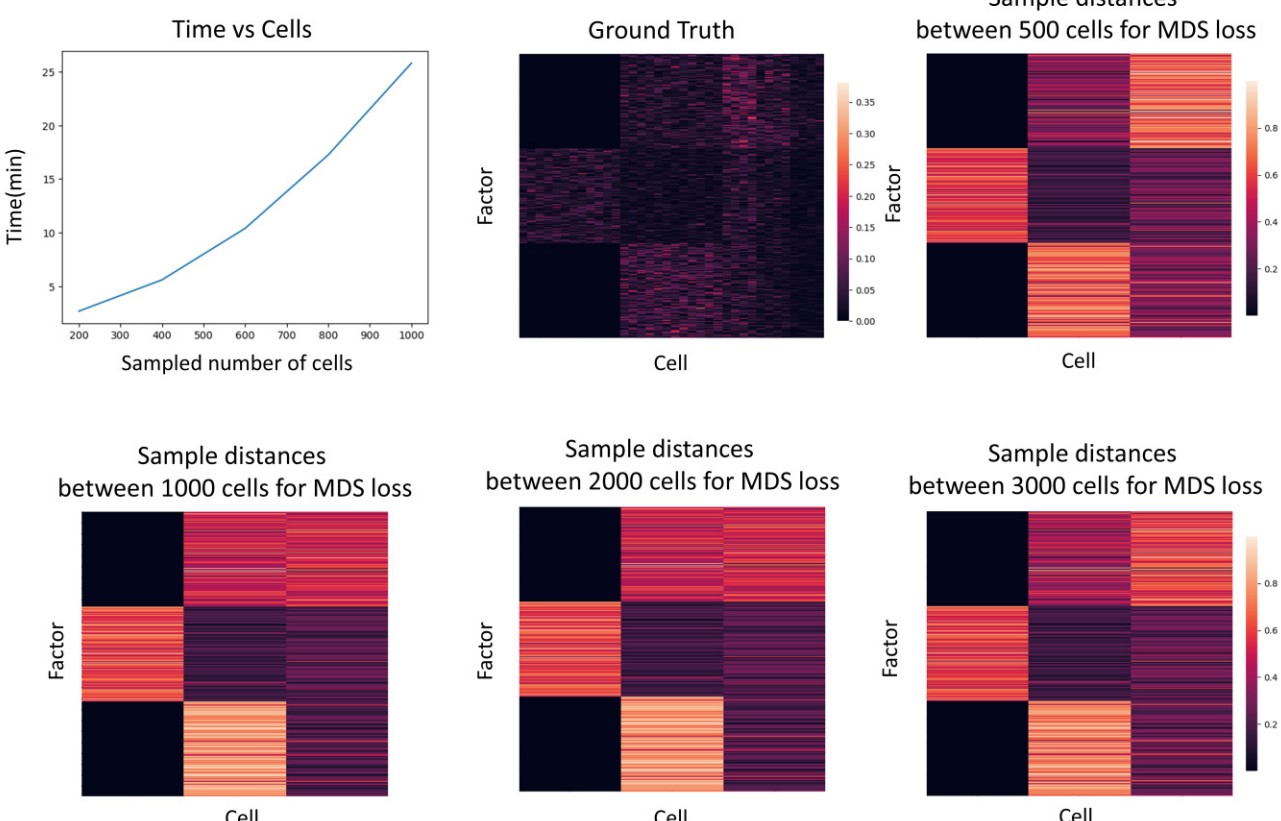

*Figure 7.* **Computational scaling of InfoGlobe with increasing cell number.** The runtime curve summarizes how computational cost changes as the number of cells increases. The heatmaps compare the reference factor structure with factor patterns recovered from progressively larger subsets of cells used for MDS-based optimization. Across sampling sizes, the learned representations preserve the major block-wise factor organization, with larger subsets producing more stable and faithful factor patterns. These results indicate that InfoGlobe maintains robust factor recovery while scaling to larger cell populations.

## G. Benchmarking Metrics Introduction

### G.1. Local and Global Structure Preservation

We evaluate structure preservation from both local and global perspectives. Let $X = \{\mathbf{x}_i\}_{i=1}^n$ denote the original high-dimensional data and $Y = \{\mathbf{y}_i\}_{i=1}^n$ denote the learned low-dimensional embedding. For each cell $i$, let $N_i^X(k)$ and $N_i^Y(k)$ be its $k$-nearest-neighbor sets in $X$ and $Y$, respectively.

**Trustworthiness.** Trustworthiness measures whether neighbors introduced in the low-dimensional embedding are also close in the original space. It penalizes false neighbors that appear in $Y$ but are not among the $k$-nearest neighbors in $X$:

$$T(k) = 1 - \frac{2}{nk(2n - 3k - 1)} \sum_{i=1}^{n} \sum_{j \in N_i^Y(k) \setminus N_i^X(k)} (r_X(i, j) - k),$$

where $r_X(i, j)$ is the rank of sample $j$ among the neighbors of sample $i$ in the original space. A higher trustworthiness score indicates fewer artificial neighbors in the embedding.

**Continuity.** Continuity measures the complementary property: whether true neighbors in the original space remain close in the embedding. It penalizes original neighbors that are missing from the low-dimensional neighborhood:

$$C(k) = 1 - \frac{2}{nk(2n - 3k - 1)} \sum_{i=1}^{n} \sum_{j \in N_i^X(k) \setminus N_i^Y(k)} (r_Y(i, j) - k),$$

where $r_Y(i, j)$ is the rank of sample $j$ relative to sample $i$ in the embedding space. A higher continuity score indicates better preservation of original local neighborhoods.

**Spearman Correlation.** To assess global structure preservation, we compute the Spearman rank correlation between pairwise distances in the original space and those in the learned embedding. Let $d_X(i, j)$ and $d_Y(i, j)$ denote the pairwise distances between samples $i$ and $j$ in $X$ and $Y$, respectively. The Spearman correlation is defined as

$$\rho = \mathrm{corr}\left(\mathrm{rank}\{d_X(i, j)\}_{i<j}, \mathrm{rank}\{d_Y(i, j)\}_{i<j}\right).$$

This metric evaluates whether the relative ordering of inter-cell distances is preserved globally. A higher value indicates better preservation of the global geometry of the data manifold.

### G.2. Clustering and Annotation Accuracy

We evaluate clustering and annotation consistency by comparing the predicted cluster labels with the reference cell-type annotations. Let $U = \{U_1, \ldots, U_R\}$ denote the clustering results produced by a method, and let $V = \{V_1, \ldots, V_C\}$ denote the reference annotations. Let $n_{rc} = |U_r \cap V_c|$, $a_r = \sum_c n_{rc}$, $b_c = \sum_r n_{rc}$, and $n$ be the total number of cells.

**Adjusted Rand Index (ARI).** ARI measures pairwise agreement between predicted clusters and reference labels while correcting for chance:

$$\mathrm{ARI} = \frac{\sum_{r,c} \binom{n_{rc}}{2} - \frac{\sum_r \binom{a_r}{2} \sum_c \binom{b_c}{2}}{\binom{n}{2}}}{\frac{1}{2}\left[\sum_r \binom{a_r}{2} + \sum_c \binom{b_c}{2}\right] - \frac{\sum_r \binom{a_r}{2} \sum_c \binom{b_c}{2}}{\binom{n}{2}}}.$$

A higher ARI indicates stronger agreement between predicted clusters and reference annotations.

**Normalized Mutual Information (NMI).** NMI measures the amount of shared information between clustering assignments and reference labels:

$$\mathrm{NMI}(U, V) = \frac{2I(U; V)}{H(U) + H(V)},$$

where

$$I(U; V) = \sum_{r,c} \frac{n_{rc}}{n} \log \frac{n \, n_{rc}}{a_r b_c},$$

and $H(U)$, $H(V)$ are the entropies of the predicted clusters and reference labels. Higher NMI indicates better alignment between clusters and annotations.

**Adjusted Mutual Information (AMI).** AMI is a chance-adjusted version of mutual information:

$$\mathrm{AMI}(U, V) = \frac{I(U; V) - \mathbb{E}[I(U; V)]}{\frac{1}{2}(H(U) + H(V)) - \mathbb{E}[I(U; V)]}.$$

By correcting for the expected mutual information under random assignments, AMI provides a more conservative measure of clustering agreement. Higher AMI indicates better annotation consistency beyond chance.

**V-measure.** V-measure is the harmonic mean of homogeneity and completeness:

$$\mathrm{V\text{-}measure} = \frac{2hc}{h + c},$$

where

$$h = 1 - \frac{H(V|U)}{H(V)}, \qquad c = 1 - \frac{H(U|V)}{H(U)}.$$

Homogeneity measures whether each cluster contains cells from a single reference class, while completeness measures whether cells from the same reference class are assigned to the same cluster. A higher V-measure indicates better simultaneous homogeneity and completeness.

**Fowlkes–Mallows Index (FMI).** FMI evaluates pairwise clustering accuracy as the geometric mean of pairwise precision and recall:

$$\text{FMI} = \sqrt{\frac{\text{TP}}{\text{TP} + \text{FP}} \cdot \frac{\text{TP}}{\text{TP} + \text{FN}}},$$

where TP is the number of cell pairs assigned to the same cluster and sharing the same reference label, FP is the number of pairs assigned to the same cluster but having different reference labels, and FN is the number of pairs assigned to different clusters but sharing the same reference label. A higher FMI indicates better pairwise agreement with the reference annotation.

# H. Additional Results

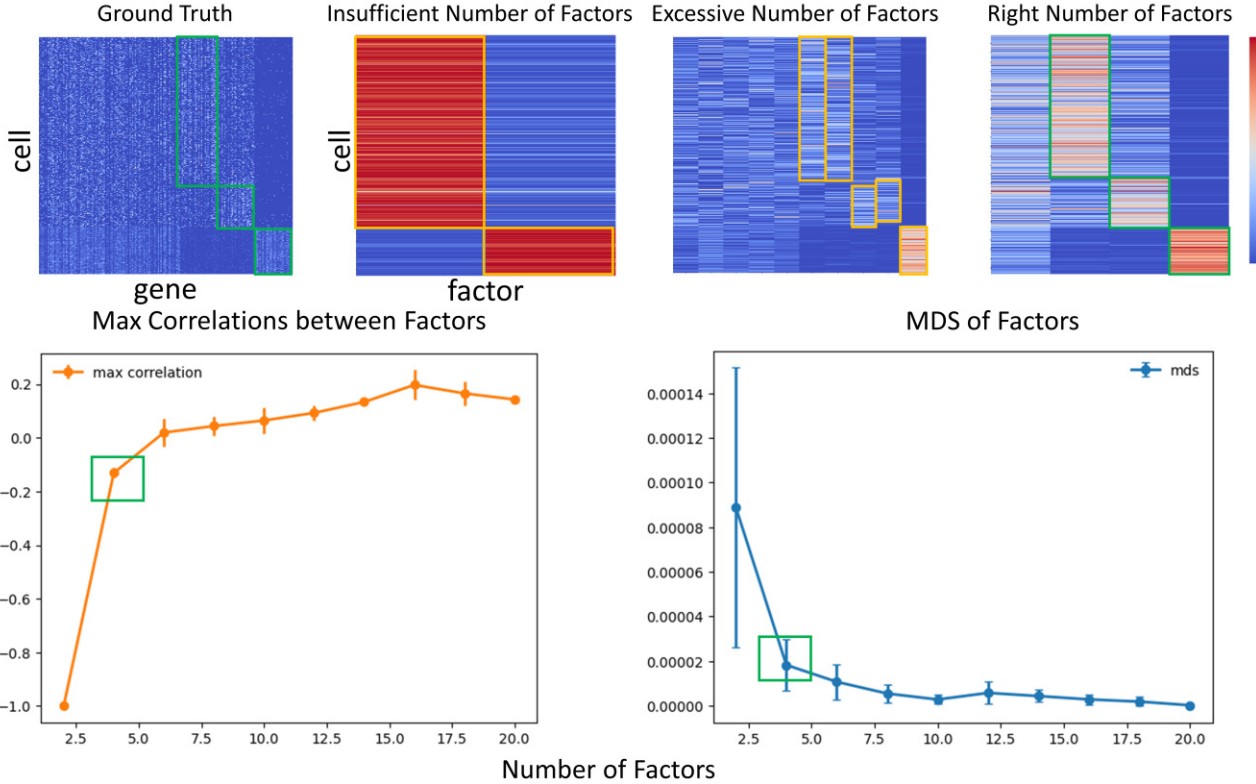

*Figure 8.* **Finding the optimal factors in InfoGlobe.** In this simulation, we simulated 4 factors, including 3 cell types that express genes from 4 factors; 3 functional factors and 1 housekeeping factor. **Top**: Illustrations of factor patterns recovered by InfoGlobe at different factor counts: insufficient factors, excessive factors and the right number of factors. When there is exessive factors, we observe the emergence of correlated factors which redundantly splits one factor into multiple sub-factors. These sub-factors share strong correlations (yellow boxes). **Bottom:** Max correlation between factors in cells and MDS loss across different factor number. When the factor number exceeds the right number of factors, we observe sudden spikes in max factor correlation and saturation in MDS loss reduction, which indicates the emergence of factor splitting and halts the iteration.

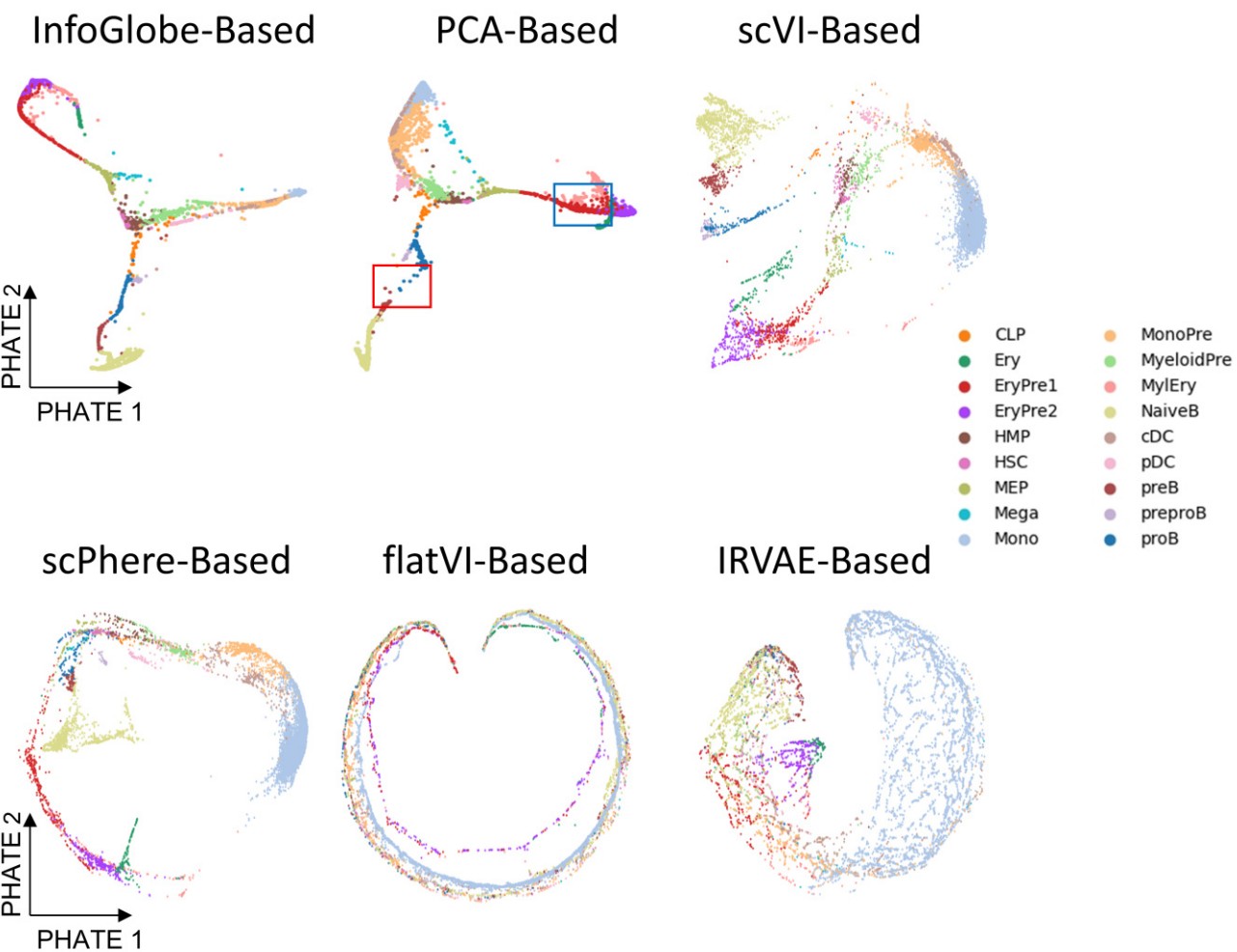

*Figure 9.* **InfoGlobe better preserves the global developmental topology in PHATE visualization.** PHATE embeddings derived from the low-dimensional representations learned by different methods are compared. InfoGlobe recovers a structured branching manifold with clear lineage continuity and biologically meaningful separation among annotated cell populations. By contrast, baseline methods show substantial geometric distortion, including local mixing, branch compression, circular collapse, and disrupted global organization. Representative regions of local mixing in the PCA-based embedding are highlighted by colored boxes. Colored points indicate annotated cell populations.

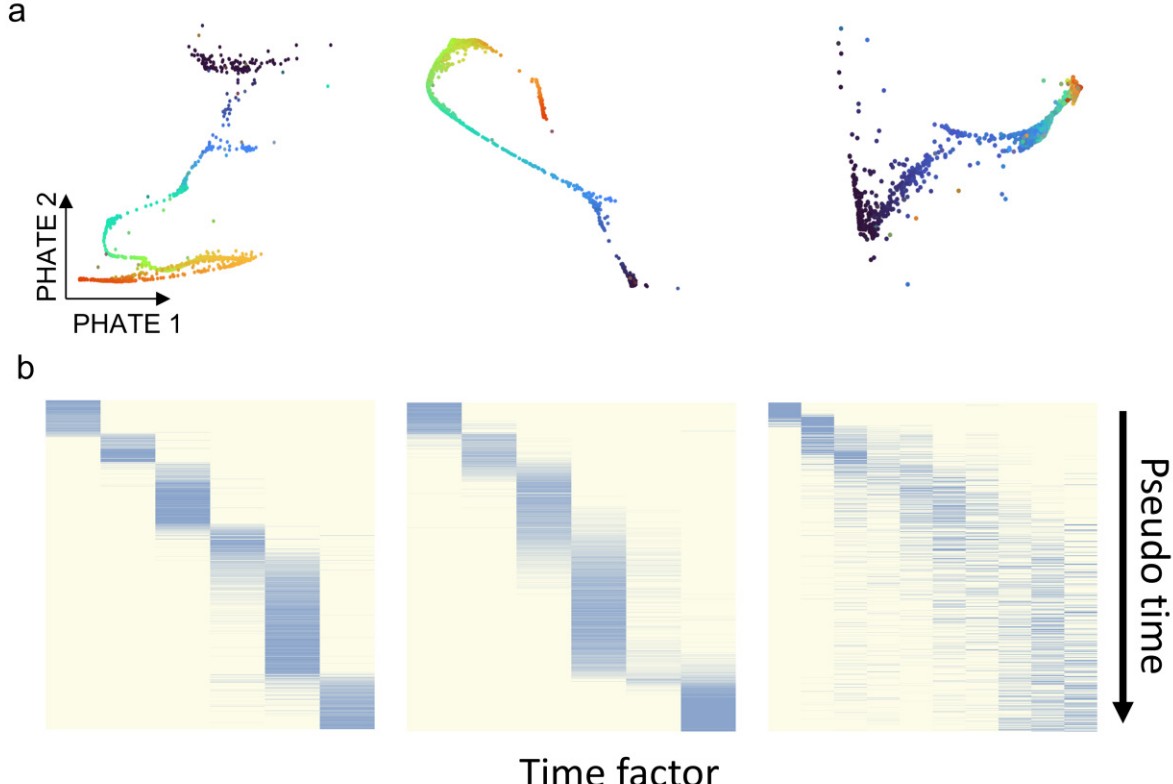

*Figure 10.* **Lineage-Specific Temporal Programs Revealed by InfoGlobe.** (a) PHATE of B cell lineage(left), PHATE of erythroid cell lineage(mid), PHATE of monocyte cell lineage(right) (b) Cell-Factor heatmap of 3 cell lineages. Each row represents a cell, while each column corresponds to a time-related factor identified by InfoGlobe. From top to bottom, cells are ordered by increasing pseudotime. The coupling between factor patterns and pseudotime progression demonstrates their biological relevance.

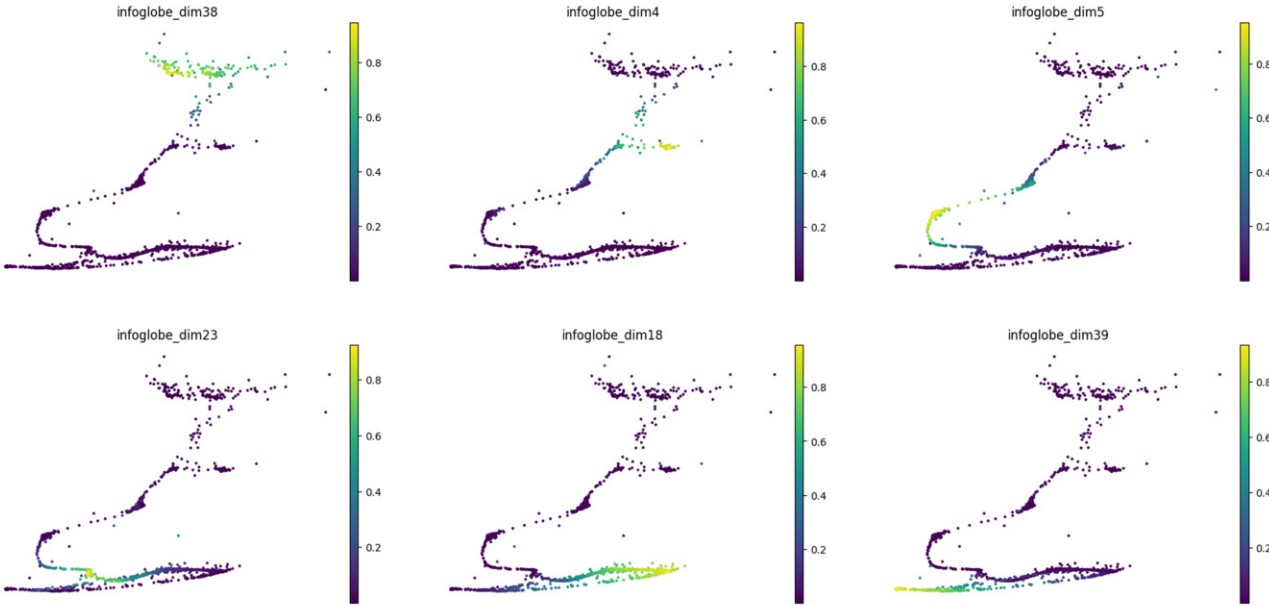

*Figure 11.* **UMAP visualization of representative time-correlated InfoGlobe factors in the erythroid lineage.** Colors denote normalized factor strength for each factor. The spatially localized and sequential activation patterns indicate that InfoGlobe captures stage-specific transcriptional programs along erythroid differentiation.

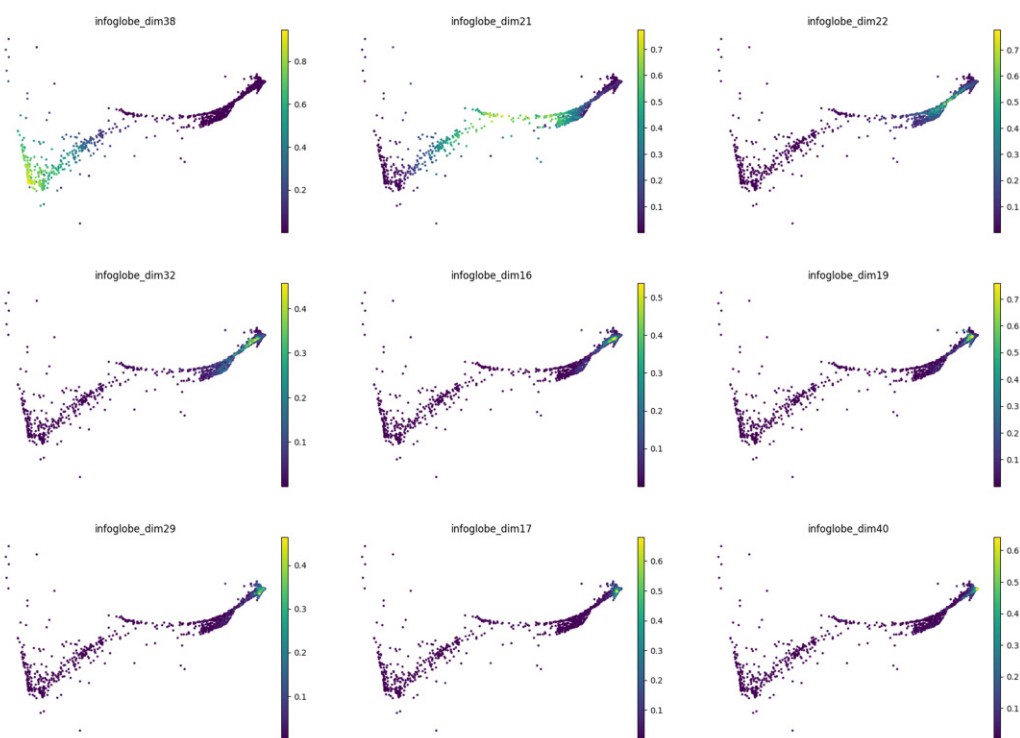

*Figure 12.* **UMAP visualization of representative time-correlated InfoGlobe factors along the B cell lineage.** Each panel shows the activity of one learned factor projected onto the B cell embedding, with colors indicating normalized factor strength. The factors display distinct activation regions along the trajectory, highlighting stage-associated transcriptional programs captured by InfoGlobe during B cell differentiation.

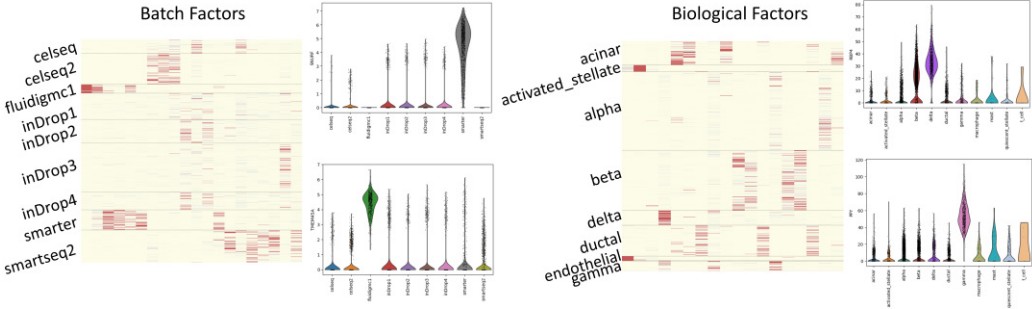

*Figure 13.* **InfoGlobe identifies distinct factors associated with batches (left) and cell types (right).** Heatmaps summarize the selective enrichment of inferred factors across batches and cell populations. Representative high-loading genes support factor interpretation: TMEM45A and SNURF mark batch-associated factors, whereas RBP4 (retinol-binding protein 4), a pancreas-associated gene, and PPY (pancreatic polypeptide), a canonical gamma-cell marker, support the biological interpretation of cell-type-associated factors.

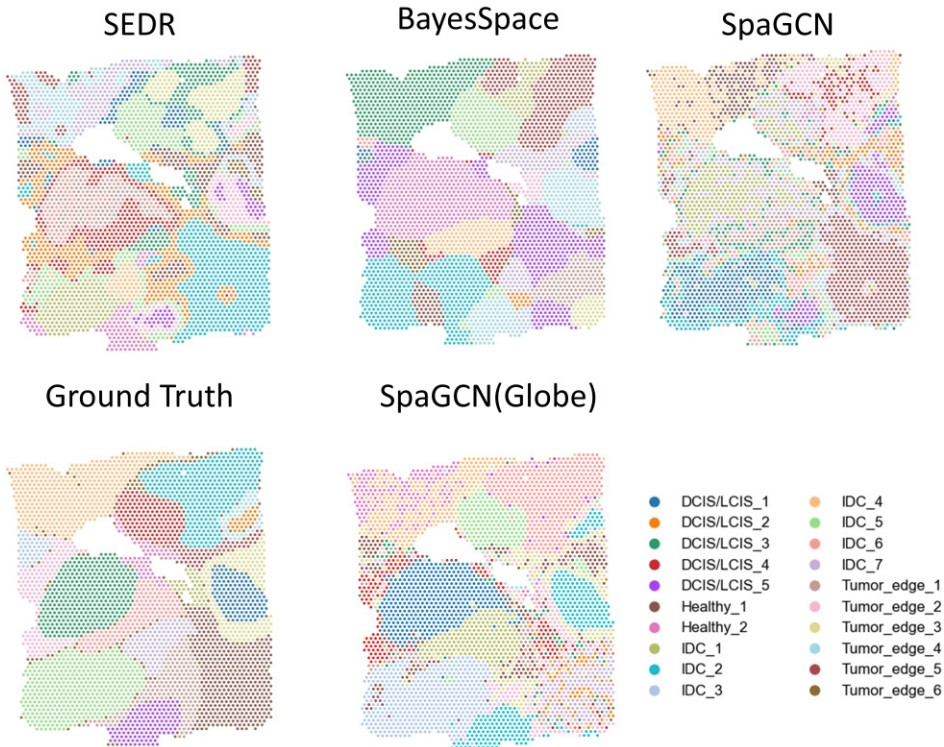

*Figure 14.* **Spatial domain identification results from different methods.** Compared with SEDR, BayesSpace, and the original SpaGCN, SpaGCN with InfoGlobe-derived representations better recovers spatially coherent domains and shows stronger agreement with the ground-truth tissue architecture. These results suggest that InfoGlobe provides a more informative representation for downstream spatial domain segmentation.

