# OpenReview forum: "InfoGlobe: Local-and-Global Information-Preserving Statistical Manifold Learning for Single-Cell Transcriptomics"
_ICML.cc/2026/Conference — ICML 2026 regular_

### Official Review · Reviewer_MzYp · 2026-03-04

**Soundness:** 3
**Presentation:** 4
**Significance:** 3
**Originality:** 3
**Overall Recommendation:** 4
**Confidence:** 2

**Summary:**

This paper proposes InfoGlobe, a geometry-preserving dimension reduction framework for single-cell transcriptomics grounded in information geometry. The method models normalized gene expression profiles as Multinomial distributions and leverages the Fisher–Rao metric to map them onto a hypersphere via square-root embedding. The approach is theoretically motivated by the equivalence between the Fisher information metric and hyperspherical arc distance, and by an extension of Chentsov’s theorem to nested Multinomial structures. Empirical evaluations on simulations, trajectory inference, batch integration, and spatial transcriptomics suggest improved global structure preservation and interpretable factor discovery.

**Compliance With Llm Reviewing Policy:**

Affirmed.

**Final Justification:**

After considering the rebuttal and subsequent discussion, I find the core claims and empirical results to be reliable and meaningful. I therefore maintain my original recommendation.

**Key Questions For Authors:**

1.	How does InfoGlobe scale computationally when the number of cells becomes very large?
2.	Does the geometry-preserving advantage remain when overdispersed count models are used instead of pure Multinomial assumptions?
3.	How robust is the greedy factor selection procedure when factors are highly correlated?
4.	How sensitive is the method to violations of the Multinomial assumption in real datasets with overdispersion?

**Limitations:**

yes

**Strengths And Weaknesses:**

Strengths
1.	The geometric interpretation of Multinomial distributions via square-root embedding onto hyperspheres is mathematically sound.
2.	The method integrates reconstruction and global geodesic preservation into a single objective.
3.	The experiments cover simulation, real scRNA-seq, and spatial transcriptomics datasets.
4.	The reported gains in global distance preservation and clustering metrics are consistent.

 Weaknesses
1.	The modeling assumption that gene expression follows an ideal Multinomial distribution may be overly simplified for real scRNA-seq data with overdispersion.
2.	The non-overlapping factor assumption used in theoretical justification is restrictive and may not reflect realistic gene program interactions.
3.	The geometric preservation term involves pairwise relationships and may limit scalability for very large datasets.

---

> ### Author Rebuttal · Authors · 2026-03-31
>
> We are immensely grateful for the acknowledgement by reviewer MzYp. We are encouraged by the reviewer for recognizing the contribution of InfoGlobe and the benefit of using Fisher-Rao as a metric for geomety-preserving dimension reduction.
>
> Q1. Can the multinomial distribution modeling account for the over-dispersion phenomenon in scRNA-seq?
>
> We strongly agree with the reviewer that scRNA-seq is over-dispersed. A key strength of InfoGlobe is its robustness in accomodating the over-dispersion effects. Over-dispersion, or dropouts, is frequently used to describe the excessive zeros in scRNA-seq data. From a gene perspective, count models are supplemented with a zero inflation term (e.g., ZINB). From a cell perspective, it is a little bit more complicated. As pointed out in "Droplet scRNA-seq is not zero-inflated" by Valentine Svensson in Nature Methods (2020), when purified to a homogeneous cell population, basic negative bionomial models contain sufficient statistics to account for all zeros. The abundant zeros in scRNA-seq can then be interpreted as the combination of: 1, the gene is only expressed in a subset of cells (zero-inflation term) and 2, for cells that express the gene, it might not be captured (negative binomial term).
>
> InfoGlobe inherits this cell-wise perspective of zero inflation: negative bionomial is sufficient for a pure cell population. Multinomial and Negative Bionomial models are inherently connected. For large trials (total UMI) and small success rate (a gene), Negative Binomial and Binomial are assymptotically the same. When extended to multi-genes, the Binomial extends to Multinomial. The Multinomial distribution in InfoGlobe models the stochastic sequencing process rather than the true gene expression proportions. Thus, the raw Multinomial parameters themselves cannot distinguish no-expression zeros and expression-but-no-capture zeros.
>
> This is exactly why we use InfoGlobe factorization. By identifying the gene-to-factor relationship, a no-capture zero in one gene can be compensated by positive expression of its sibling genes.
>
> To reflect the robustness of InfoGlobe against over-dispersion, we have conducted an experiment evaluating the factorization efficacy under various sequencing depths. At low sequencing depths, there is exessive over-dispersion. The results are summarized in Fig. 1. As shown in the figure, the factorization results by InfoGlobe are rather consistent across various sequencing depths, proving that it can withstand high over-dispersions.
>
> [Fig1. InfoGlobe factorization is stable across various dropout intensities.](https://shorturl.at/Qn80a)
>
> To summarize, the design of InfoGlobe takes the high-dropout and overdispersion phenomenon in scRNA-seq into consideration. The main difference from ZINB models is that it takes a cell-centric perspective rather than a gene-centric perspective. The per-cell multinomial model and the per-gene ZINB model are inherently connected. By merging genes into factors, the stochasticity in genes cancels out. Experiments demonstrate the robuestness of InfoGlobe against sequencing depth variations and intensified over-dispersion.
>
> Q2. Is the non-overlapping factor assumption too restrictive?
>
> This is a very insightful question. InfoGlobe does not forbid overlapping factors. The non-overlapping factor assumption, however, is important for the Chentzov theorem, which establishes geometric isometry between the original gene manifold and the low-dimensional factor manifold.
>
> While being an essential part of the Chentsov theorem, InfoGlobe does not incorporate this restriction. Fig. 2 shows a 3-factor simulation where genes overlap between factors. The results prove that InfoGlobe can correctly attribute genes to multiple factors in agreement with ground truth.
>
> [Fig2. InfoGlobe supports overlapping gene-to-factor mapping.](https://shorturl.at/oaVQw)
>
> Q3. How does InfoGlobe scale when the number of cells becomes very large?
>
> This is a valid concern. The computation of InfoGlobe can be broken down into 2 blocks: 1, measuring the reconstruction loss and recording the gradients; 2, measuring the MDS loss and recording the gradients. For reconstruction loss, the complexity is $O(N\*F\*G) $, with $N$ cells, $F$ factors and $G$ genes. For the MDS loss computation, the complexity is $O(P\*(G+F)) $, with $P$ the number of cell pairs used for distance regularization.
>
> To improve scalability, we can subsample the cell pairs involved in MDS loss computation. Fig. 3 reports the InfoGlobe factorization results with respective cell pair sub-sampling intensity. As shown in the figure, the factorization by InfoGlobe remains stable and accurate, proving that sub-sampling cells for pairwise MDS regularization is a viable option.
>
> [Fig3. InfoGlobe factorization remains stable after sub-sampling cell pairs for MDS regularization.](https://shorturl.at/UpOY9)
>
> Due to length limit, we refer our answer about factor numbers to our response to nFwN.

---

> > ### Author Rebuttal · Reviewer_MzYp · 2026-04-02
> >
> > Thank you for the detailed and thoughtful rebuttal.  While I still have some concerns regarding computational complexity on very large datasets, these do not significantly affect my overall assessment. My evaluation remains unchanged.

---

> > > ### Author Response · Authors · 2026-04-06
> > >
> > > Dear Reviewer MzYp:
> > >
> > > We sincerely appreciate your thoughtful evaluation and support, and we will continue to further explore the issue of computational complexity on very large datasets in our future work.

---

### Official Review · Reviewer_Rq4g · 2026-03-06

**Soundness:** 3
**Presentation:** 3
**Significance:** 3
**Originality:** 3
**Overall Recommendation:** 4
**Confidence:** 5

**Summary:**

This paper provides a new embedding method for cells using Fisher-Rao metric using scRNA-seq. The writing is good. My main concern is that the topic of geometry-preserving dimension reduction for single-cell data is relatively mature, and the paper mainly introduces a new metric formulation rather than a new problem setting. So the contribution of this paper is marginal. Moreover, given the existence of many other embedding methods (such as Laplacian Eigenmap, UMAP,...), what is the advantage of the proposed method compared with them? And what is the rationale behind the current selection of comparison methods? Why were these methods chosen instead of other embedding approaches?

**Compliance With Llm Reviewing Policy:**

Affirmed.

**Final Justification:**

I decide to keep the score.

**Key Questions For Authors:**

This paper provides a new embedding method for cells using Fisher-Rao metric using scRNA-seq. The writing is good. My main concern is that the topic of geometry-preserving dimension reduction for single-cell data is relatively mature, and the paper mainly introduces a new metric formulation rather than a new problem setting. So the contribution of this paper is marginal. Moreover, given the existence of many other embedding methods (such as Laplacian Eigenmap, UMAP,...), what is the advantage of the proposed method compared with them? And what is the rationale behind the current selection of comparison methods? Why were these methods chosen instead of other embedding approaches?

**Limitations:**

This paper provides a new embedding method for cells using Fisher-Rao metric using scRNA-seq. The writing is good. My main concern is that the topic of geometry-preserving dimension reduction for single-cell data is relatively mature, and the paper mainly introduces a new metric formulation rather than a new problem setting. So the contribution of this paper is marginal. Moreover, given the existence of many other embedding methods (such as Laplacian Eigenmap, UMAP,...), what is the advantage of the proposed method compared with them? And what is the rationale behind the current selection of comparison methods? Why were these methods chosen instead of other embedding approaches?

**Strengths And Weaknesses:**

This paper provides a new embedding method for cells using Fisher-Rao metric using scRNA-seq. The writing is good. My main concern is that the topic of geometry-preserving dimension reduction for single-cell data is relatively mature, and the paper mainly introduces a new metric formulation rather than a new problem setting. So the contribution of this paper is marginal. Moreover, given the existence of many other embedding methods (such as Laplacian Eigenmap, UMAP,...), what is the advantage of the proposed method compared with them? And what is the rationale behind the current selection of comparison methods? Why were these methods chosen instead of other embedding approaches?

---

> ### Author Rebuttal · Authors · 2026-03-31
>
> We are immensely grateful for the comments by reviewer Rq4g and the appreciation for adopting the Fisher-Raw metric framework.
>
> We are deeply sorry that our writing leaves the impression that our work is "yet another geometric dimension reduction but with a different metric". We are grateful for having the opportunity to clarify our regretfully confusing writing.
>
> First, while UMAP and t-SNE are indeed geometric preserving, they only preserve the local but not global geometry. We believe this is a critical yet often overlooked problem. Many of the manifold learning algorithms, including UMAP and t-SNE, are built based on the locally Euclidean property of Reimannian geometry. That is, on a Remannian manifold, the geodesic distance between remote points is intractable, while distances between nearby points are asymptotically Euclidean. Following this observation, many of the "geometric preserving" manifold learning algorithms opted for the strategy of preserving local geometry measured by Euclidean distance, while forfeiting the accurate characterization of distances between remote pairs.
>
> InfoGlobe differs from these locally geometry-preserving methods in that it seeks to faithfully preserve both the local and global geometry. This is the first major difference.
> [Fig.1 Stability of global distance](https://shorturl.at/hDPBb)
>
> Second, the low-dimensional manifold learnt by InfoGlobe is a geodesic manifold. Part of the reason that UMAP and t-SNE forfeit the accurate characterization of long distances is that the geodesic long distance themselves are intractable. In scRNA-seq, gene expressions are first normalized to a constant total transcriptomic size and then log-transformed to amplify fold change differences. This generates a $M:=\sum_i e^{x_i} = C$ manifold on which cell resides. For more than two dimensions, computing the geodesic distance on this log transformed manifold is NP-hard. Thus, many algorithms use Euclidean distance as a replacement, which cuts through the manifold rather than adhering to the surface. Consequently, the distance is no longer geodesic, which relinquishes the need for their accurate portrayal in the low dimensional manifold.
>
> InfoGlobe on the other hand, is an isometric geodesic dimension reduction. That means the low dimensional manifold learnt by InfoGlobe is perfectly embedded on the hyper-surface of the high dimensional manifold and the Remainnian distance (arch distance) between points in the low dimensional manifold, is indeed the geodesic distances in the original manifold, for both local and distant points. This gives InfoGlobe the unique ability to establish a stable global scaffold between cells. This is illustrated in Fig 2.
>
> [Fig 2. InfoGlobe produces a geodesic manifold](https://shorturl.at/QcmQn)
>
> Third, InfoGlobe jointly characterizes on/off effects and less/more effects. Log-transformation emphasizes fold changes at the cost of accepting severe deformation at low expression values. Let $\delta$ be the expression difference of a gene between two cells. The difference after transformation would be $\Delta = log(\frac{o+X+\delta}{o+X})$, with $o$ being the offset and $X$ the baseline. When $o$ is sufficiently small,$\Delta \rightarrow \infty$ as $X \rightarrow0$ and $\Delta \rightarrow 0$ as $X \rightarrow \infty$. Consequently, log transformation prioritizes on/off effects at the expense of less/more effects.
>
> With the square-root transformation, InfoGlobe strikes a good balance between both effects. The distance difference $\Delta = \sqrt{X+\delta} -\sqrt X)$ decreases at a much slower state than log transformation. This means that InfoGlobe moderately enhances the on/off differences without completely overwhelming the less/more differences (Fig 3).
>
> [Fig 3.  Harmonization of on/off and less/more effects](https://shorturl.at/bVqts)
>
> Finally, from a factorization perspective, InfoGlobe establishes an organic connection between gene expression and factor strength, which does not involve any feature selection (such as highly variable genes) and is stable across sequencing depths.
>
> By modeling gene expression and factor expression as Multinomial distributions, InfoGlobe establishes a probabilistically meaningful connection that faithfully describes the sampling process with a hierarchical Multinomial sampling process. By circumventing the log-transformation, InfoGlobe is not troubled by the severe distortions at low-expression values, which enhances its factorization stablity. The factorization stability is summarized in Fig. 4.
>
> [Fig 4. InfoGlobe factorization is stable across various dropout intensities.](https://shorturl.at/Qn80a)
>
> In summary, InfoGlobe addresses a critical problem in dimension reduction where global distances cannot be accurately characterized and that the low-dimensional manifold is not a geodesic submanifold. It also provides stable and transparent factorization across sequencing depths. We are happy to discuss further with the reviewer.

---

> > ### Author Rebuttal · Reviewer_Rq4g · 2026-04-01
> >
> > Thanks for your response. I decide to keep the score.

---

> > > ### Author Response · Authors · 2026-04-06
> > >
> > > Dear Reviewer Rq4g:
> > >
> > > We deeply appreciate your careful assessment and your encouraging response to our revision.
> > > Your valuable feedback has helped us improve the manuscript, and we are sincerely thankful for your support.

---

### Official Review · Reviewer_nFwN · 2026-03-11

**Soundness:** 3
**Presentation:** 3
**Significance:** 3
**Originality:** 3
**Overall Recommendation:** 4
**Confidence:** 4

**Summary:**

This paper proposes InfoGlobe, a dimension reduction method designed for single-cell transcriptomics motivated by information geometry. In particular, the authors show that the Fisher-Rao metric on the gene-group manifold is isometric to gene manifolds, and design InfoGlobe to preserve both local and global geometry.

**Compliance With Llm Reviewing Policy:**

Affirmed.

**Final Justification:**

The authors addressed my concerns so I raised by score from 3 to 4.

**Key Questions For Authors:**

In the anonymous github website, I didn't find any readme file or instruction about how to implement it, reproduce the results in the manuscript, or use it in practice.

See also "weakness" for more questions.

I am open to reevaluate this paper if the weaknesses are addressed during the rebuttal phase.

**Limitations:**

Only one limitation is discussed, i.e., the gene-to-factor mapping is globally constant.

**Strengths And Weaknesses:**

**Strength**

The work is motivated by both an important practical problem (dimension reduction for single-cell transcriptomics) and a rigorous mathematical observation (Fisher-Raw metric), leading to a novel method with strong empirical performance. I personally appreciate this type of work and enjoyed reading it.

The math intuition is simple but clean. The entire journey from math to method is clearly explained and well written, easy to follow.

Downstream tasks are pretty comprehensive. Given that dimension reduction is unsupervised, there is no gold standard to show that the embeddings are "good". However, the authors tried hard to make it convincing that the embedding from InfoGlobe can benefit a bunch of downstream tasks including global and global structure preservation, clustering, factor analysis, trajectory analysis, batch correction, and spatial domain detection.



**Weakness**

How to choose the latent dimension is not discussed enough. This is a tough question in general, but competitors like PCA does admit a clean way to choose the dimension (explained variance). What about the proposed method? For practitioners without technical background, how to choose this dimension is crucial for potential impact of this work.

The clustering performance reported in Table 2 might due to the choice of Leiden algorithm, instead of the high quality of embeddings. It will be more convincing to at least show similar trend using another clustering method.

The ST experiment compares BayesSpace, SEDR, SpaGCN using PCA, and SpaGCN using InfoGlobe. However, since the purpose is to show InfoGlobe generates better embeddings, it will be more convincing to compare BayesSpace using PCA vs BayesSpace using InfoGlobe, SEDR using PCA vs SEDR using InfoGlobe, SpaGCN using PCA vs SpaGCN using InfoGlobe. Such comparison will better support the quality of InfoGlobe embeddings, while comparing different domain detection methods for ST is not really the focus of this paper.

The competitors chosen in each experiments are not consistent. For example, scPHERE, scVI, FlatVI and IRVAE are only used in the simulation study. Why they are not compared in the trajectory analysis or spatial domain detection tasks? For instance, what if we use SpaGCN based on scVI embeddings?

There is a rich literature in batch correction, but none of them is considered in Figure 5.

Line 255 (left), fisher-rao should be Fisher-Rao.

Line 262 (left), the $d_FR$ should be $d_{\text{FR}}$.

Line 330 (right), the reference Figure 6B should be Figure 4C.

The figure on Page 19 is too big, can't read the final panel or the caption. In fact, all figures and captions in the appendix are not in the same qualify with figures in the main paper. I strongly recommend the authors to pay equal attention to them as they are also a key component of this manuscript (otherwise why the authors include them).

---

> ### Author Rebuttal · Authors · 2026-03-30
>
> We are greatly encouraged by the reviewer for appreciating our work. We are genuinely thrilled by the deep insights that the reviewer grasped from our work. We apologize for the hustle in parts of the result section and we deeply agree with the reviewer that any supplementary material should share the same quality as the main text. Once again, we extend our apology to all reviewers. We have completely overhauled the supplementary sections (will appear). We have also added the readme of our github repo for improved accessibility (sorry for the major blunder).
>
> Q1. How to determine the number of factors?
>
> This is a great question. InfoGlobe is different from PCA in the sense that as the number of factors changes, the gene-factor and cell-factor could also change at the same time, although not necessarily to the detriment of separating distinct cell populations or measuring their distances in the low dimensional manifold.
>
> To illustrate why this is the case, and subsequently address this issue, we propose a simplest example. Let there be only two gene programs $F_1$ and $F_2$, in which cells vary in the expression ratios. Let $F_1$ encompass genes $F1 \rightarrow \{g_1,g_2,g_3,g_4\}$ and $F2 \rightarrow \{g_5,g_6,g_7,g_8\}$. The true degree of freedom of cells in terms of factor is 2.
>
> With only 1 factor, InfoGlobe lacks sufficient manifold flexibility, causing large mismatches between high- and low-dimensional Fisher–Rao distances.
>
> At 2 factors, InfoGlobe correctly recovers the low-dimensional manifold, gene-to-factor weights, and factor-to-cell matrix, yielding maximal agreement between low- and high-dimensional inter-cell distances and thus low Riemannian MDS loss.
>
> Starting at 3 factors, InfoGlobe splits factors into sub-factors, e.g., $F1$ into $F1a \rightarrow \{g_1,g_2\}$ and $F1b \rightarrow \{g_3,g_4\}$.Although redundant, this does not alter the geometry or inter-cell distances, as the ratio $F1a:F1b=C$ is constant.
>
> To summarize, when there is insufficient number of factors, we would expect a large MDS loss. At the exact right number of factors, we would expect a low MDS loss with minimum (even negative) Pearson correlations between factor strengths. When there is excessive number of factors, we would envision prevalent factor splitting, creating a large cohort of linearly correlated factors among cells.
>
> Following this observation, we propose a factor number detection workflow. Briefly, we would sweep the factor count. We would execute InfoGlobe and keep track of the MDS loss as well as the max correlation among factors in cells. When the MDS loss saturates and when we observe a sudden spike in max inter-factor correlation, we know that we have just exceeded the right factor count and the loop terminates. Fig 1. illustrates this strategy in action.[Fig.1 Find optimal number of factors](https://shorturl.at/3sUMM)
>
> Q2. How to evaluate the quality of the embeddings?
>
> Due to length limits, we refer much of our reply to the full response to reviewer fwpR. Briefly, in scRNA-seq, because of the intensified stretch at low expression values by log transformation, PCA (or NMF) is incapable of simultaneously incorporating both on/off expression effects and less/more expression effects. In addition, PCA, log transformed or not, is extremely sensitive to the excessive inclusion of housekeeping genes.
>
> These effects are summarized in Fig 2 below. InfoGlobe, on the other hand, maintains good separation.[Fig.2 Embedding evaluation](https://shorturl.at/RSM3Q)
>
> For the clustering comparison, we added K-means, DBSCAN, and Louvain results in Figure 3, which show a similar trend. [Fig.3 Clustering Benchmark correction](https://shorturl.at/BCMqg)
>
> Q3. Expanding the ST experiment and tarjectory analysis to additional methods and maintaining consistency in benchmarks.
>
> We apologize for our oversight. They have since been corrected as shown in Fig 4 and 5 below.
> [Fig.4 ST Benchmark correction](https://shorturl.at/5u9tP)
> [Fig.5 Trajectory Benchmark](https://shorturl.at/eNljD)
>
> Q4. Consider additional methods for batch correction.
>
> We apologize for misguiding the audience. InfoGlobe does not seek to replace or compete against batch integration algorithms. Instead, we observe that without purposed regression, with just factorization, InfoGlobe attributes genes to biological factors and batch/sample factors. We also observe that by simply cropping out the batch factors, cells between different platforms becomes naturally aligned. Fig 6 demonstrates this effect.[Fig.6 Biological and batch factors](https://shorturl.at/MfSID)
>
> However, alignment after removing batch factors is still imperfect, suggesting residual biological differences. For better integration, InfoGlobe should be combined with batch correction methods. A comparison with batch integration algorithms is shown in Fig. 7.[Fig.7 Batch correction benchmark](https://shorturl.at/HZqYh)
>
> Other comments are addressed accordingly. They are skipped here due to length limit.

---

> > ### Author Rebuttal · Reviewer_nFwN · 2026-04-03
> >
> > Thank the authors for addressing my comments. I increased my score.

---

> > > ### Author Response · Authors · 2026-04-06
> > >
> > > Dear Reviewer nFwN:
> > >
> > > Thank you again for your positive feedback and for recognizing our efforts in addressing your concerns. We are truly grateful for your thoughtful evaluation and kind support of our work.

---

### Official Review · Reviewer_fwpR · 2026-03-12

**Soundness:** 2
**Presentation:** 2
**Significance:** 3
**Originality:** 3
**Overall Recommendation:** 4
**Confidence:** 4

**Summary:**

This paper presents a method for dimensionality reduction of scRNAseq datasets.

They model the data measurement process with multinomial distribution and define a factorization of the distribution into a hierarchy of two-level multinomial distributions: “factors” following a multinomial distribution and genes following a multinomial distribution within each factor. The factors can be interpreted as gene programs or technical variations.

They formally define the dimensionality reduction process as a linear factorization of the data matrix, optimizing the reconstruction and distance preserving. They define the distance using Fisher-Rao metric on the statistical simplex of multinomial distributions.

Furthermore, they proposed a way to rank the factors by variance explained using a greedy algorithm.

The method was benchmarked on real and synthetic datasets against latest methods, and showed promising performance on embedding metrics as well as downstream tasks (trajectory inference, batch correction, spatial segmentation).

**Compliance With Llm Reviewing Policy:**

Affirmed.

**Final Justification:**

The rebuttal has addressed my concerns in clarifications on key formula, metrics, hyperparameter details, and relationship with existing methods. Thus, I have raised my score.

**Key Questions For Authors:**

1. The method is a linear dimensionality reduction method, which is worse at preserving information than nonlinear methods like t-SNE, UMAP, Diffusion Maps, PHATE, especially when the dimensionality is low. Can you provide details on the number of factor used, how well they preserve the data information (e.g. the reconstruction loss), and how that number is chosen?
2. In equation (8), the distance preserving loss sums the differences between distances in original space and factor space, rather than absolute or squared difference. This is problematic because it can be reduced if all the distances between factors are maximized, regardless of the reference distance in data space. Can you justify this design choice or clarify if there is a typo.
3. Need clarification on the metics used in section 5.1. What is “continuity” and why does it measure global distance preserving? What is Spearman correlation computed on and what does it measure?

**Limitations:**

Yes

**Strengths And Weaknesses:**

**Soundness**

The method is generally technically sound, with rigorous theory of statistical distributions and information geometry. The idea makes sense and is appropriate for the problem. However, there are some details that needs to be clarified (see questions) on the design of the loss function and the evaluation metrics.

**Presentation**

The presentation is clear, especially the related work and background. There are some issues in the method description, for example, there was an abrupt jump from section 4.2 to 4.3, which can cause confusion on what is the method trying to optimize. It could be improved with an overall description of the method pipeline or some pseudocode. The abuse of notation X in 4.3 is also confusing. The details on the optimization process is also missing, although source code is provided. The evaluation metrics were not explained in the result section, which needs clarification.

**Significance**

The dimensionality reduction problem in scRNAseq is important, and the downstream applications shown in the paper are impactful.

**Originality**

The combination of multinomial distribution, Fisher-Rao metric and factorization for single-cell data dimensionality reduction is novel.

---

> ### Author Rebuttal · Authors · 2026-03-30
>
> We are immensely grateful for the comments from reviewer fwpR.
>
> InfoGlobe packs two contributions: 1, Fisher-Rao metric and the square-root transformation for transcriptomic similarity measurement; and 2, Information-preserving factor analysis. InfoGlobe does not compete with UMAP, t-SNE, DiffusionMap and Phate. Instead, InfoGlobe has good synergy with these popular manifold learning methods. Due to the high dimensionality and sparsity in scRNA-seq, these manifold learning algorithms typically are not applied directly to raw data. Instead, they are applied to the top PC space extracted from PCA. InfoGlobe replaces PCA when applied in tandem with these manifold learning algorithms.
>
> InfoGlobe seeks to fix a critical, but often overlooked flaw in PCA or NMF: the low dimensional subspace they produce is typically not geodesically embedded ihigh-dimensional cellular manifold. This is important as a key premise of dimension reduction is that while the gene space is very high dimensional, cells are actually distributed in a low-dimensional or low-degree-of-freedom subspace manifold, that is perfectly embedded within the high-dimensional manifold. Geometrically, we can visualize it as cells are distributed on a hypersurface in a 30k-dimensional gene expression space, but on that hypersurface, there is a perfectly geodesic sub-hypersurface that is everywhere tangent to the original hypersurface, which sufficiently captures the distribution pattern of cells. Dimension reduction thus seeks to extract that low-dimensional hypersurface and characterize cells by its low-dimensional factor coordinates.
>
> Without special design, PCA and NMF project cells onto a low-dimensional submanifold that cuts through, rather than lies on, the original cellular manifold. In scRNA-seq, after library-size normalization and log transformation, cells reside on the manifold $M:=\sum_i e^{x_i} = C$. As Fig 1 shows, PCA projects cells to a straight line that cuts through this manifold rather than a straight line that lies on $M$. Because of this, the geodesic distance (Eucledian) between two points in the low-dimensional PC space, even if there is no loss in compression, differs from the geodesic distance in $M$.
>
> InfoGlobe addresses this critical flaw by projecting cells to a hypersphere through square-root transformation. This projection reflects the Fisher-Rao information distance between Multinomials. The low-dimensional manifold of InfoGlobe is a geodesic submanifold that is everywhere tangent to $M$. The geodesic distance of the low-dimensional manifold is exactly the geodesic distance of $M$ (Fig 1).
>
> [Fig.1 InfoGlobe produces a geodesic manifold](https://shorturl.at/QcmQn)
>
> InfoGlobe provides 3 benefits:
>
> A, balanced characterization of on/off and less/more effects. Log transformation emphasizes fold changes but severely distorts low-expression differences. For a gene with baseline expression$X$ and difference $\delta$ between two cells, the transformed difference is $\Delta = log(\frac{o+X+\delta}{o+X})$, where $o$ is the offset. When $o$ is small, $\Delta \rightarrow \infty$ as $X \rightarrow 0$ and $\Delta \rightarrow 0$ as $X \rightarrow \infty$. Thus, log transformation prioritizes on/off effects at the expense of less/more effects (Fig 2). InfoGlobe strikes a good balance between the two.
>
> [Fig.2 Harmonization of on/off and less/more effects](https://shorturl.at/bVqts)
>
> B, reliable global distance measurement. UMAP and t-SNE prioritizes the representation of local neighborhoods thus the global distance between cells fluctuates. Phate and DiffusionMap assign arbitrary large distances between disconnected subgraphs, thus does not preserve a global scaffold between various terminally differentiated cell types. PCA, because of its cut-through projection, is highly sensitive to sequencing depth and distorts inter-cell-type distances nonlinearly (Fig 3). InfoGlobe produces more consistent global distances across sequencing depths.
>
> [Fig.3 Stability of global distance](https://shorturl.at/hDPBb)
>
> C, reliable factor analysis. The linear model design of InfoGlobe is designed specifically for factor analysis. For visualization, we recommend applying UMAP on the substrate of InfoGlobe. For the same reason as in B, the factors produced by InfoGlobe are more stable (Fig4).
>
> [Fig.4 Factor stability](https://shorturl.at/hDxnr)
>
> We are immensely grateful for pointing out unclarity in our writing. Equation 8 has been revised to: $L_{\text{geom}}=\sum_{i,j}(d_{\text{FR}}(x_{\cdot i}, x_{\cdot j})-d_{\text{FR}}(z_{\cdot i},z_{\cdot j}))^2$. And Continuity is measuring whether true neighbors in the original space remain neighbors in the embedding, thus reflecting local preservation. Spearman correlation is computed between pairwise cell-cell distances from each learned embedding and the ground-truth factor space, measuring global distance-rank preservation.
>
> For the number of factors, due to length limit, we refer to our response to nFwN.

---

> > ### Author Rebuttal · Reviewer_fwpR · 2026-04-02
> >
> > Thank you for the rebuttal!
> > My concerns have been addressed and I have raised the score.

---

> > > ### Author Response · Authors · 2026-04-06
> > >
> > > Dear Reviewer fwpR:
> > > We sincerely appreciate the time and effort you have devoted to reviewing our manuscript. Your insightful and constructive comments have greatly enhanced the clarity and rigor of our work.

---

### Decision · Program_Chairs · 2026-04-30

**Decision:**

Accept (regular)

**Comment:**

This paper introduces a new information geometric framework for dimensionality reduction in single-cell transcriptomics that is useful for preserving local and global distances. This framework can be used as input or a preprocessing step for further dimensionality reduction, similar to the role PCA typically plays.

After the rebuttal and discussion period, all reviewers lean towards accepting this paper. None of the reviewers have lingering, major concerns with the paper. Overall, the paper is technically sound, well-written, and makes an important contribution to those in the ICML community that work with single-cell transcriptomics data. The authors should make sure to incorporate all of the new results in the camera ready version of the paper.